# Convergent evolution of aerobic fermentation through divergent mechanisms acting on key shared glycolytic genes

Linda C Horianopoulos [1,2]✉, Antonis Rokas [3,4] & Chris Todd Hittinger [1]✉

## Abstract

As the tree of life becomes increasingly accessible to molecular investigations, describing mechanisms underlying evolutionary convergence and constraint will be crucial to understanding diversification. The lineage including the model yeast *Saccharomyces cerevisiae* evolved aerobic fermentation in part through an ancient whole genome duplication and retention of glycolytic genes. To evaluate glycolytic rates across diverse yeasts, we developed and deployed an extracellular acidification rates (ECAR) assay on 299 species that span more than 400 million years of evolution and identified a clade in the genus *Saturnispora* that convergently evolved aerobic fermentation. Through comparative genomics and transcriptomics, we found that several glycolytic genes had higher expression and novel cis-regulatory elements in aerobically fermenting *Saturnispora* species. When the transcription factor required for their activation was deleted in *Saturnispora dispora*, the mutants had reduced glycolytic rates and increased respiration. Intriguingly, many of the upregulated genes are orthologous to duplicated glycolytic genes in *S. cerevisiae*. These divergent genetic mechanisms affecting the same set of genes suggest that there are strong evolutionary constraints on how aerobic fermentation can arise.

Subject Categories Evolution & Ecology; Metabolism; Microbiology, Virology & Host Pathogen Interaction

## Introduction

The convergent evolution of complex traits provides insights into the mechanisms and constraints that govern their repeated emergence. Convergence can arise through distinct or common genetic mechanisms. One of the most compelling examples of convergent evolution in a complex metabolic trait is the evolution of C4 photosynthesis in more than 60 independent lineages (Sage et al, 2011). C4 plants have repeatedly and independently evolved mechanisms to increase photosynthesis efficiency by partitioning biochemical reactions and allowing the key photosynthesis enzyme, RuBisCO, to be in a high $CO_2$ environment (Brown et al, 2011; Williams et al, 2012; Lyu et al, 2025). Comparisons across C4 grasses have identified multiple underlying evolutionary mechanisms, including gene family expansions of key enzymes, parallel amino acid changes, and similar cis-regulatory elements in the promoters of C4 enzymes (Lyu et al, 2024). For example, a specific comparison between rice (C3) and sorghum (C4) revealed that the promoters driving the C4 enzymes in sorghum co-opted an ancestral cis-regulatory element for cell type specificity, thereby promoting this physical partitioning of biochemical reactions (Swift et al, 2024). These innovations highlight how improved metabolic efficiency has been rediscovered in many independent lineages through different mechanisms altering the abundance, expression patterns, or activity of the same key C4 enzymes.

Within the yeasts of the subphylum Saccharomycotina, the Crabtree/Warburg Effect, in which yeasts ferment even in the presence of oxygen, is a complex metabolic trait that has evolved in at least two independent lineages: in *Saccharomyces* and relatives and some *Brettanomyces* species (Rozpedowska et al, 2011). Comparative phenotyping of up to forty species demonstrated that the Crabtree-positive phenotype likely evolved in a stepwise fashion and that overflow metabolism, wherein cells maximize energy production rates through glycolysis but produce fermentation products, is the fundamental mechanism driving aerobic fermentation (Hagman and Piškur, 2015; Hagman et al, 2013). Some yeasts, including *S. cerevisiae*, have undergone dramatic genetic changes, including preferential retention of duplicated glycolytic genes from an ancient whole genome duplication (WGD), further increasing their fermentative capacity (Conant and Wolfe, 2007; Escalera-Fanjul et al, 2019). *Brettanomyces* species, which are also Crabtree-positive, do not have the high glucose consumption rates of *S. cerevisiae*, but they do repress respiration when glucose is present (Hagman et al, 2013; Rozpedowska et al, 2011). These independent occurrences of aerobic fermentation suggest that there may be additional lineages of yeasts with independently evolved preferences for fermentation.

[1]Laboratory of Genetics, J. F. Crow Institute for the Study of Evolution, Center for Genomic Science Innovation, DOE Great Lakes Bioenergy Research Center, Wisconsin Energy Institute, University of Wisconsin-Madison, Madison, WI, USA. [2]Department of Food Science, University of Guelph, Guelph, ON, Canada. [3]Department of Biological Sciences, Vanderbilt University, Nashville, TN, USA. [4]Evolutionary Studies Initiative, Vanderbilt University, Nashville, TN, USA. ✉E-mail: lhoriano@uoguelph.ca; cthittinger@wisc.edu

Complex traits, such as the Crabtree/Warburg Effect, are difficult to assess using high-throughput assays. However, since a high glycolytic rate and overflow metabolism are thought to precede the evolution of the Crabtree/Warburg Effect, we developed an assay to evaluate the glycolytic rate of diverse species based on glucose-induced extracellular acidification rates (ECAR). ECAR has been used to study mammalian cells, primarily cancer cells, which display the Crabtree/Warburg Effect wherein they accumulate lactate as a byproduct of rapid aerobic glycolysis (Schmidt et al, 2021; Zhang and Zhang, 2019). Acidification measurements have also been applied in *S. cerevisiae* to assess fermentative capacity in industrial settings (Sigler, 2013; Opekarova and Sigler, 1982; Yamashoji et al, 2020; Sigler et al, 1981). Acidification occurs in yeast during glucose assimilation as neutral glucose is taken up, phosphorylated, and converted into charged glycolytic intermediates, organic acids, and carbon dioxide. The intracellular accumulation of charged intermediates and organic acids lowers the pH of the cytosol, but the cell must keep the cytosol buffered to ensure proper enzymatic functions. Thus, the cell extrudes protons, which rapidly lowers the extracellular pH upon glucose addition (Kotyk et al, 2003; Orij et al, 2011, 2009). At later culture stages, the extracellular accumulation of organic acids can further lower the extracellular pH (Sigler and Hofer, 1991). More recently, ECAR was measured with the Seahorse Extracellular Flux analyzer to assess glycolytic rate in pathogenic yeasts (Tucey et al, 2018; Yoo et al, 2024; Zhang et al, 2018). Assessing glycolytic rates across a wide range of species could potentially uncover novel instances of high glycolytic rates and provide insight into the genetic mechanisms underlying high overflow metabolism. Identification of species with high glycolytic rates may also have important implications as increasing efforts are being made to use non-conventional yeasts for biotechnological applications (Geijer et al, 2022; Wang et al, 2025; Radecka et al, 2015; Pyne et al, 2023; Patra et al, 2021).

Herein, we developed a microtiter plate-based assay to screen diverse yeast species for ECAR. We confirmed that the species in the order Saccharomycetales, including the model yeast *S. cerevisiae*, display rapid ECAR phenotypes, which are consistent with their rapid glycolytic rates. However, we also identify multiple species in the genus *Saturnispora*, including *Saturnispora dispora*, that have rapid ECAR compared to other species in the order Pichiales. This order already contains two yeasts of considerable biotechnological interest: *Komagataella phaffii* (previously known as *Pichia pastoris*), which is a leading chassis for protein production (Heistinger et al, 2020; Gasser and Mattanovich, 2018), and *Pichia kudriavzevii* (also known as *Issatchenkia orientails*, *Candida glycerinogenes*, and *Candida krusei*), which is emerging as an important microbial cell factory for the production of organic acids (Tan et al, 2025; Suthers and Maranas, 2022). Furthermore, additional species within the genus *Pichia* are actively being developed for organic acid production (Pyne et al, 2023). This emerging interest in the industrial applications of Pichiales warrants further work to understand the metabolic capabilities of this order to guide their development as microbial cell factories. Moreover, characterizing the genetic mechanism driving the strong fermentative capacity in *Saturnispora* can reveal the evolutionary constraints (if any) on rapid fermentation and guide rational strategies to maximize rates of organic acid production in Pichiales.

To understand how the unusually high glycolytic rate of *Sat. dispora* and its relatives evolved, we used comparative genomics and transcriptomics and found that several genes encoding glycolytic enzymes had increased expression at the transcript level in species with a rapid ECAR phenotype. Intriguingly, genes orthologous to those whose duplications are key to the high glycolytic rate of *S. cerevisiae* also had higher expression in the rapid ECAR *Saturnispora*, which prompted us to investigate the promoter sequences of glycolytic genes for binding sites of transcription factors (TFs) known to regulate carbon metabolism. We found that there were more predicted Gal4p-binding sites ($CGGN_{11}CCG$) in the promoters of some glycolytic genes in the rapid ECAR species. Although Gal4p is named for its role in galactose metabolism in *S. cerevisiae* (Johnston, 1987), this role is evolutionarily derived and specific to Saccharomycetales (Harrison et al, 2022). Gal4p homologs play important roles in regulating carbon metabolism in *Candida albicans* (Askew et al, 2009) and glycolysis in *K. phaffii*, although these species do not natively exhibit aerobic fermentation (Ata et al, 2018). Using novel genetic tools that we developed for *Sat. dispora*, we showed that *GAL4* was necessary for the induction of the most highly expressed glycolytic gene, *TDH*. Altogether, we found an independently evolved group of yeasts with a high glycolytic rate phenotype achieved through the recruitment of cis-regulatory elements driving the direct transcriptional upregulation of orthologs of the same glycolytic genes that retain duplicates in *S. cerevisiae*. This convergence suggests that the evolution of increased glycolytic rate is constrained and requires high activity of key glycolytic enzymes, which can be achieved by distinct molecular mechanisms.

# Results

## Rapid ECAR in *Saccharomyces* and relatives and in the distantly related genus *Saturnispora*

We developed a high-throughput method to measure the glucose-dependent ECAR of yeast species as a proxy for glycolytic rate using a 96-well plate and a standard microplate reader (Fig. EV1). We plotted the mean of three biological replicates per species as a color gradient across the yeast phylogeny to identify clades with similar ECAR profiles. Accordingly, we found that many species in the order Saccharomycetales, particularly those yeasts that underwent an ancestral whole genome duplication (WGD), such as *S. cerevisiae*, had rapid acidification (Fig. 1A; Dataset EV1). We also compared our ECAR data with parameters associated with the Crabtree/Warburg Effect using a previously collected dataset primarily composed of Saccharomycetales yeasts (Hagman et al, 2013). We found significant positive correlations between ECAR and the oxygen consumed, as well as biomass yield, and a negative correlation between ECAR and ethanol yield (Fig. EV2; Dataset EV2). Since rapid acidification has a negative ECAR value, the rapid ECAR phenotype is thus associated with low oxygen consumption, low biomass yield, and high ethanol yield. Although this result was expected and validated the ability of our method to capture glycolytic rate, we were particularly interested in other clades that showed a rapid ECAR phenotype. For example, we were surprised to find a group in the order Pichiales in the genus *Saturnispora* that showed a rapid ECAR phenotype (as low as −0.032 pH/h in *Sat. dispora*). These species have not been thoroughly characterized but are in the same order as multiple biotechnologically relevant species, including the Crabtree-negative species *K. pastoris* (low ECAR = 0.022 pH/h). The growing biotechnological interest in Pichiales yeasts prompted our specific interest in the metabolism of these species and in the evolution of the rapid ECAR trait in some *Saturnispora* species.

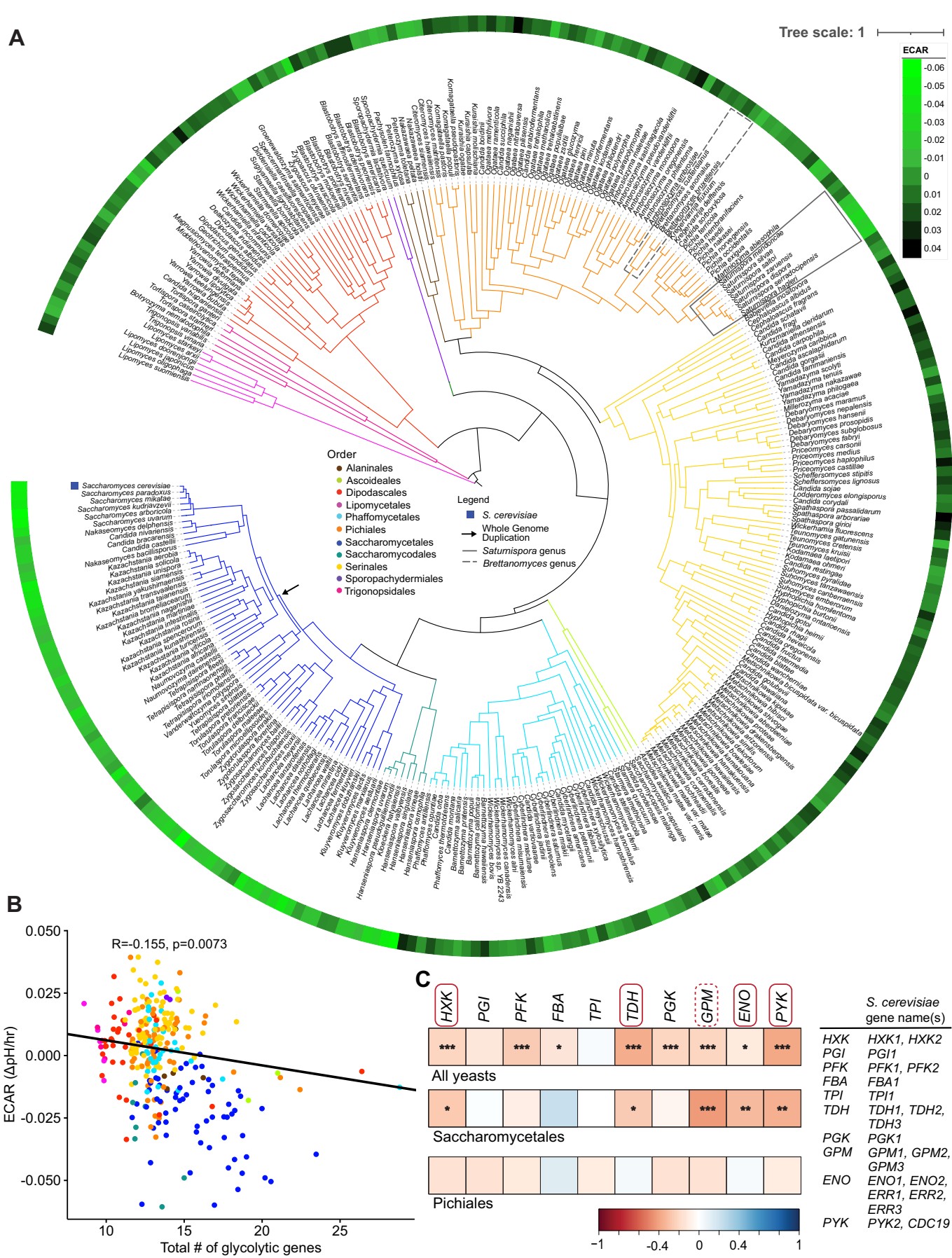

**Figure 1.   Variation of extracellular acidification rates (ECAR) across the yeast subphylum is associated with glycolytic genes.**

(A) The mean of the measured ECAR for each species is visualized as a circle around the yeast phylogeny, with the lighter green blocks corresponding to more rapid ECAR. The model yeast *S. cerevisiae*, the whole genome duplication (WGD) in the Saccharomycetales (arrow), the genus *Saturnispora* (box), and the genus *Brettanomyces* (dashed box) are highlighted on the phylogeny. (B) A scatter plot showing the correlation between ECAR and the total number of glycolytic genes where each dot represents a single species. The correlation and *P* values are based on a phylogenetic generalized least squares model, which accounts for correlation stemming from shared phylogenetic history. (C) Pearson correlations between the number of copies of each glycolytic gene and the measured ECAR. The gene names in boxes have retained duplicates in *S. cerevisiae*; solid boxes indicate duplicates retained in all post-WGD yeasts, whereas the dashed boxes indicate duplicates retained in some but not all post-WGD yeasts (Conant and Wolfe, 2007). Significance of the Pearson correlation is indicated as *$P < 0.05$, **$P < 0.01$, ***$P < 0.005$. The yeast phylogeny was pruned from Shen et al (Shen et al, 2018). Source data are available online for this figure.

## High glycolytic rate correlates with gene duplication in *S. cerevisiae* and relatives

The high glycolytic rate in *S. cerevisiae* is hypothesized to be, at least in part, due to an ancestral WGD and subsequent preferential retention of glycolytic genes (Conant and Wolfe, 2007; Hagman et al, 2013; Conant and Wolfe, 2008). To further validate the results of our screen and test the generality of the hypothesis that having a higher number of glycolytic genes is associated with rapid ECAR across yeasts, we compared the number of glycolytic genes and the measured ECAR from each yeast species in our dataset. We found a significant correlation between the total number of glycolytic genes and ECAR across all yeast species after correcting for phylogeny (Fig. 1B). To interrogate this pattern further, we examined the correlations between the size of the gene families encoding each enzymatic step in glycolysis and ECAR. Again, we found strong correlations when looking at all species (Fig. 1C). However, when we focused on the taxonomic orders containing rapid ECAR species, we noticed that several of these correlations held true in the order Saccharomycetales but were not significant within the order Pichiales (Fig. 1C). When the correlations of individual genes were corrected for phylogeny, we found that the significance values of their correlations were lost in the Saccharomycetales, likely due to the limited power. Thus, the correlation between ECAR and glycolytic gene family size may be driven mainly by the well-known WGD in the Saccharomycetales (Fig. EV3). Altogether, these results suggest that the *Saturnispora* species with the rapid ECAR phenotype in the order Pichiales likely evolved independently from the duplication of glycolytic genes observed in *S. cerevisiae*.

## Divergent phenotypes of rapid and low ECAR species in *Saturnispora*

To interrogate the Pichiales-specific rapid ECAR phenotype, we sought to identify closely related species with diverging phenotypes. Within the genus *Saturnispora*, we found that several species consistently showed a rapid ECAR phenotype, namely *Sat. dispora*, *Sat. hagleri*, *Sat. serradocipensis*, *Sat. zaruensis*, and *Sat. saitoi*. Conversely, *Sat. silvae* and *Sat. mendoncae* consistently displayed a low ECAR phenotype (Fig. 2A). Moving forward, we focused on four species, two rapid ECAR and the two low ECAR with the hypothesis that the rapid ECAR species would have metabolic phenotypes more similar to known Crabtree-positive yeasts. To confirm that the acidification reflected rapid glucose consumption, we measured glucose depletion in the media in a typical growth experiment. The rapid ECAR species consumed glucose more rapidly than the low ECAR species (Fig. 2B). The rapid ECAR

species even produced ethanol and acetate from overflow metabolism, despite being grown under highly aerobic conditions in baffled flasks (Fig. 2B). Accordingly, the rapid ECAR species also grew to a lower final OD compared to the low ECAR species (Fig. 2C). These divergent metabolic phenotypes show that these rapid ECAR yeasts are highly fermentative and indeed exhibit the Crabtree/Warburg Effect under aerobic conditions, which prompted us to interrogate potential genetic differences between the rapid ECAR and low ECAR species.

## Transcriptional profiling of rapid and low ECAR species

Since there were no duplications in glycolytic genes in the rapid ECAR *Saturnispora* species (Fig. 1C; Dataset EV1), we used comparative transcriptomics to evaluate differences in gene expression between the rapid and low ECAR species. To compare across species, we mapped the reads to genes in each species, generated counts, summed counts across genes in the same orthogroup, and calculated transcripts per million (TPM) for each orthogroup. Since our primary interest was the glycolytic rate trait, we specifically retrieved the orthogroups encoding central carbon metabolism genes. These data revealed that the rapid ECAR species had significantly higher expression of several glycolytic genes, especially genes encoding hexokinases and the enzymes of lower glycolysis (*TDH*, *GPM*, *ENO*, *PYK*) (Fig. 3). The rapid ECAR species also had significantly higher expression of the genes encoding the hexose transporters (*HXT*) and the first step towards alcoholic fermentation, pyruvate decarboxylase (*PDC*) (Fig. 3). These transcriptional differences are consistent with the rapid glucose consumption and aerobic fermentation phenotype. Accordingly, we also found that the low ECAR species had higher expression of genes encoding tricarboxylic acid (TCA) cycle enzymes. In particular, we noted that the expression of the gene encoding pyruvate carboxylase and multiple genes encoding components of the pyruvate dehydrogenase complex had significantly lower expression in the rapid ECAR species, which suggests that reduced respiratory capacity also contributes to overflow metabolism (Fig. EV4). The differences at the transcriptional level of both glycolysis and the TCA cycle support the conclusion that the metabolisms of these closely related yeasts are highly divergent and that low ECAR species favor aerobic respiration.

## The Gal4p transcription factor contributes to the rapid glycolytic rate in rapid ECAR *Saturnispora*

Since we observed such a striking difference in the expression of glycolytic genes between rapid and low ECAR yeasts, we

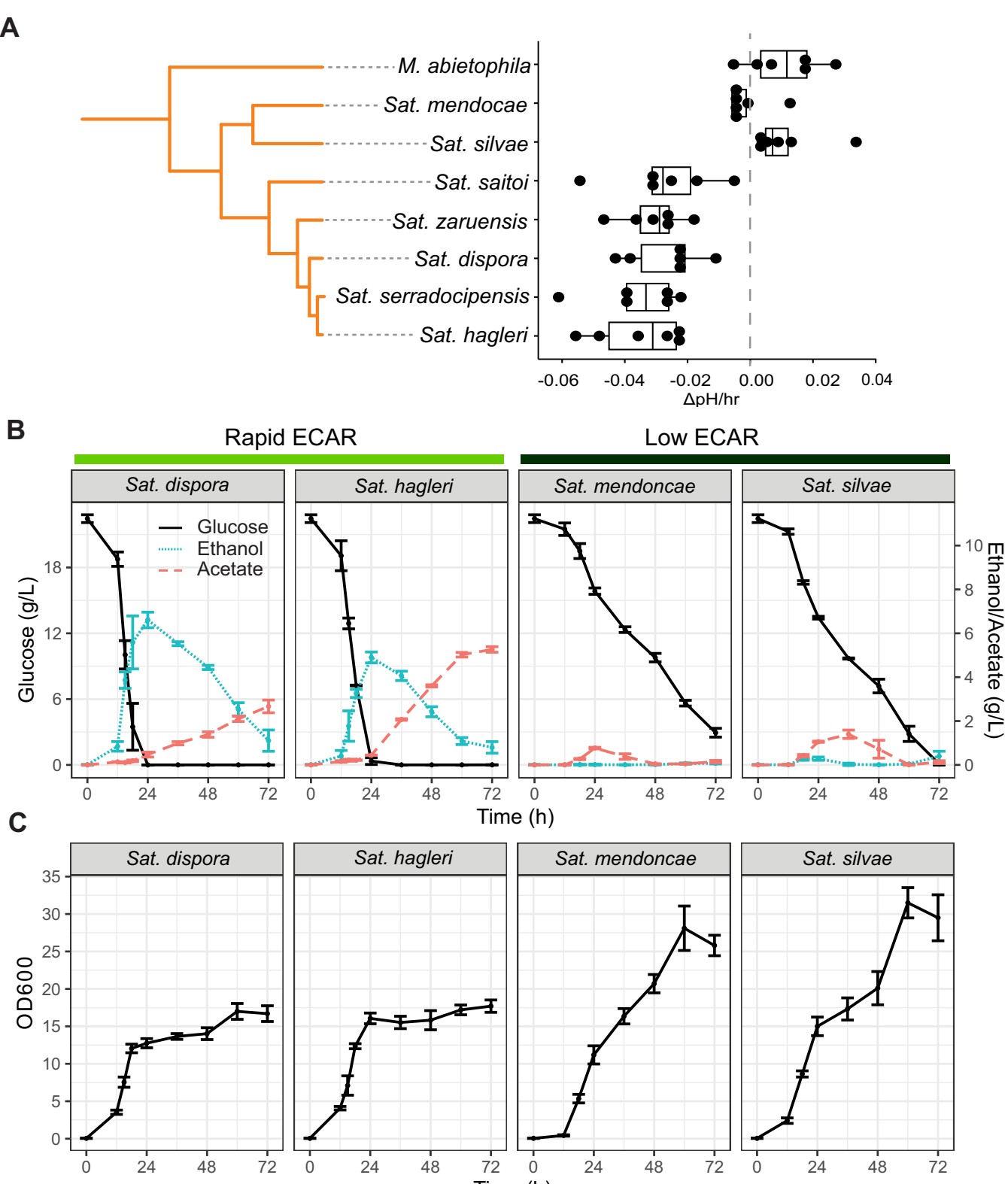

hypothesized that this difference was due to genetic differences in the regulatory regions of their glycolytic genes. To test this hypothesis, we retrieved the nucleotide sequences 1 kb upstream of each glycolytic gene as the putative promoter sequences of these

species, as well as other Pichiales for comparison. We predicted the conserved transcription factor (TF) binding sites in these putative promoters to identify TFs that had more binding sites in our rapid ECAR species compared to the low ECAR species. We found that

**Figure 2.  Species within the genus *Saturnispora* display divergent metabolic phenotypes.**

(A) ECAR measurements identified species with consistently low or consistently rapid ECAR within the genus *Saturnispora*. ECAR was also measured for *Martiniozyma abietophila* as a closely related outgroup. The center lines of boxplots represent the median, the bounds of the boxes represent the interquartile range, the whiskers represent the spread of the data, and each dot represents one of six biological replicates. The (B) extracellular metabolites in the supernatant and (C) growth of two rapid ECAR and two low ECAR species were measured during growth in a 250-ml baffled flask, which was highly aerobic. The error bars represent the standard deviation of four biological replicates, and the point represents the mean. Source data are available online for this figure.

the promoters of some glycolytic genes had additional Gal4p-binding sites in the rapid ECAR species (Fig. 4A; Dataset EV3). In rapid ECAR *Saturnispora*, there was an extra predicted Gal4p-binding site in the promoter of *TDH*, the most highly expressed glycolytic gene. There were also uniquely predicted Gal4p-binding sites in the promoters of *HXK* and *ENO* in the rapid ECAR species, which were absent from the low ECAR species (Fig. 4A; Dataset EV4). Interestingly, when we compared these promoters to all other Pichiales species assayed, we found that there was considerable variation and that *Saturnispora* spp. have more Gal4p-binding sites in the glycolytic gene promoters compared to most other Pichiales, suggesting that this regulatory rewiring may be a recent innovation.

To investigate the importance of Gal4p in the rapid ECAR phenotype, we performed a phylogenetic analysis and found that, despite some lineage-specific duplications in this orthogroup, *GAL4* is likely a 1:1 ortholog in *S. cerevisiae*, *K. phaffii*, *Sat. dispora*, and most yeasts (Fig. EV5). We then developed a system to genetically modify a rapid ECAR yeast, *Sat. dispora*, based on methods previously described for *K. phaffii* (De Schutter and Callewaert, 2012) and generated a deletion mutant by replacing this *GAL4* ortholog with a *kanMX* or *natMX* resistance cassette using long homology to direct homologous recombination. Two independent deletion mutants lacking *GAL4* had decreased glucose consumption, ethanol production, and a low ECAR phenotype (Fig. 4B–D). We also found that mutants lacking *GAL4* had a more respiratory phenotype with a higher oxygen consumption rate (Figs. 4E and EV6), further supporting the hypothesis that the Gal4p transcription factor is required for the rapid ECAR metabolic phenotype and aerobic fermentation in *Sat. dispora*.

To test whether there was a direct cis-regulatory effect of Gal4p, we also made a GFP reporter construct driven by the *Sat. dispora TDH* promoter. Furthermore, we made a version of this reporter in which the predicted Gal4p-binding sites were mutated using site-directed mutagenesis. We inserted these reporters into a genomic safe haven, in which there were no predicted genes, in both the wild-type background and the *gal4Δ::natMX* deletion mutants. The expression of GFP was significantly reduced in the wild-type background when the predicted binding sites were mutated. The expression was also significantly reduced when the reporter construct with intact Gal4p-binding sites was transformed into the *gal4Δ::natMX* mutant. Together, these results show that, in the absence of either the trans-acting transcription factor or the cis-regulatory binding sites, the promoter activity is greatly reduced (Fig. 5A–C). Although the GFP expression was very low in the *gal4Δ::natMX* deletion background, there was a small but significant difference between the wild-type and mutant reporter constructs, suggesting that the mutations may also have interfered with the ability of other transcription factors to bind. Altogether, this result directly shows that Gal4p is necessary to activate the most highly expressed glycolytic gene.

## Discussion

The discovery of shared or novel mechanisms to increase glycolytic rate can help us understand the evolutionary constraints and guide the manipulation of this trait across diverse yeast species. In this study, we developed an assay to assess variation in glycolytic rate across the yeast subphylum. Comparative genomics corroborated previous reports that duplications of glycolytic genes were associated with increased glycolytic rates in yeasts, particularly in the Saccharomycetales where a WGD occurred (Conant and Wolfe, 2007; Hagman and Piškur, 2015; Dashko et al, 2014; Hagman et al, 2013; Lin and Li, 2011). However, there were no duplications or expansions in glycolytic genes associated with the rapid ECAR phenotype in the Pichiales. This contrast suggested that the rapid ECAR phenotype we observed in several *Saturnispora* species had an independent mechanism. Using transcriptomics, we observed that several glycolytic genes were upregulated in species with rapid ECAR. While these results were consistent with the phenotypic observations, we were surprised to see how closely they mirrored the gene duplications in *S. cerevisiae* (Conant and Wolfe, 2007). In particular, the genes encoding enzymes in lower glycolysis, which were duplicated and retained in *S. cerevisiae*, had much higher expression in the rapid ECAR species. The association of elevated expression of lower glycolysis genes with aerobic fermentation is also consistent with previous evidence that pull through lower glycolysis is key to fermentative capacity (Smits et al, 2000). Furthermore, lower glycolysis carries more flux than any biochemical alternative (Court et al, 2015), suggesting that upregulating these existing genes may be the most effective strategy to increase glycolytic rate.

Since the driving force between this phenotypic difference seemed to be at the transcriptional level, we retrieved the putative promoter sequences upstream of glycolytic genes to identify transcriptional activators that had more binding sites in the promoters in rapid ECAR species. Several of the glycolytic gene promoters had more conserved binding sites for the Gal4p transcription factor. Although Gal4p is well characterized for its role in activating the Leloir pathway for galactose catabolism in *S. cerevisiae* (Johnston, 1987), it is also known to have other roles in metabolism in different yeasts (Askew et al, 2009; Kim et al, 2024; Ata et al, 2018). Indeed, in another Pichiales yeast, the protein production chassis, *K. phaffii*, overexpression of Gal4p has been shown to induce overflow metabolism (Ata et al, 2018). In the case of *Sat. dispora*, this yeast natively has a rapid glycolytic rate and produces ethanol and acetate as overflow metabolites. The promoters with additional Gal4p-binding sites in rapid ECAR species compared to the low ECAR species in *Saturnispora* were upstream of *HXK, TDH*, and *ENO*. This configuration again bore a striking parallel to modeling in *S. cerevisiae* that predicted these three enzymes have a large impact on flux when their abundance

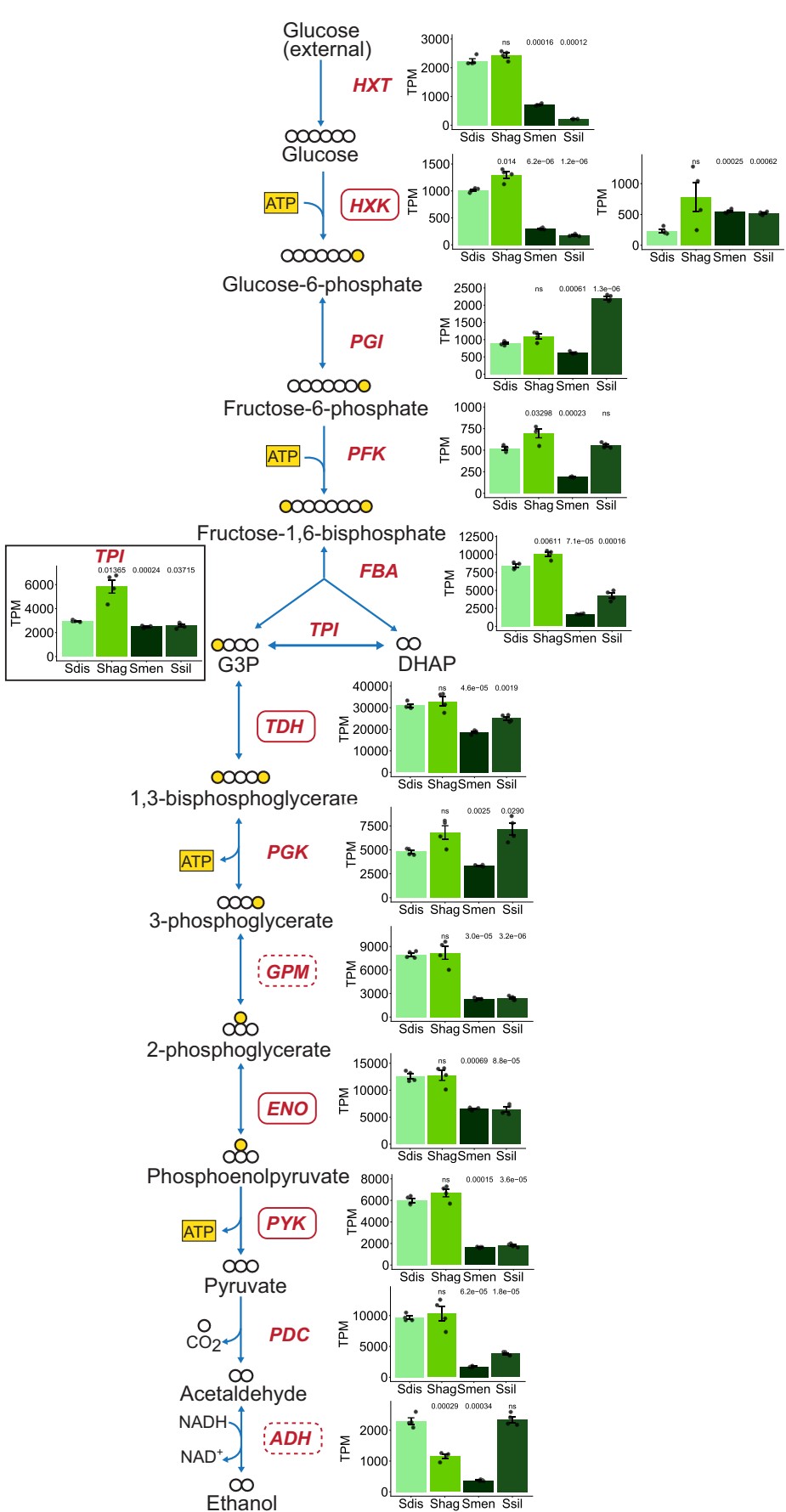

**Figure 3.  Divergent expression patterns of glycolytic genes between rapid ECAR and low ECAR *Saturnispora* species.**

The gene expression for orthologous glycolytic genes across two rapid ECAR (*Sat. dispora* Sdis, *Sat. hagleri* Shag) and two low ECAR (*Sat. mendoncae* Smen, *Sat. silvae* Ssil) are shown as transcripts per million (TPM). The bar height represents the average of four biological replicates; each replicate is shown as a black dot, and the error bars represent the standard error of the mean. The significance is relative to the TPM in *Sat. dispora* (ns not significant). In the schematic of glycolysis, carbon backbones are white, and phosphates are yellow. Boxes around gene names are orthologous to genes that have retained duplicates in *S. cerevisiae* as in Fig. 1C; solid boxes indicate duplicates retained in all post-WGD yeasts, whereas the dashed boxes indicate duplicates retained in some but not all post-WGD yeasts (Conant and Wolfe, 2007).

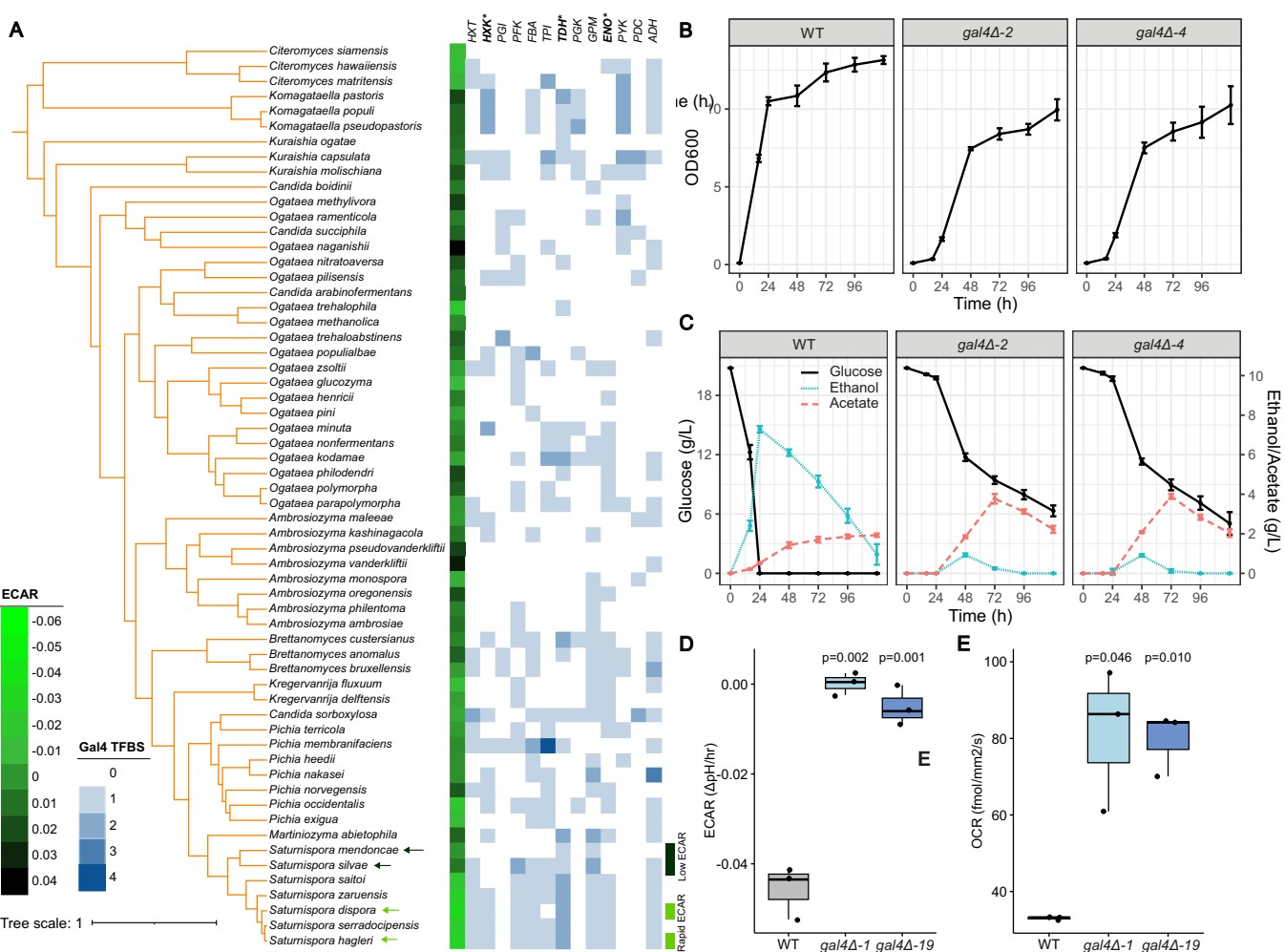

**Figure 4.  The Gal4p transcription factor contributes to the rapid ECAR metabolic phenotype in *Sat. dispora*.**

(A) The number of predicted Gal4p-binding sites in the putative promoters 1 kb upstream of each glycolytic gene in all tested Pichiales species. The genes marked with an asterisk have an additional binding site in the rapid ECAR species. The focal rapid and low ECAR species used in this study are highlighted with arrows on the phylogeny. The (B) growth and (C) extracellular metabolites for the *Sat. dispora* wild-type (WT) control and two independent *gal4Δ* deletion mutants. Two high ECAR and two low ECAR species were measured during growth in a highly aerobic 250-ml baffled flask. The error bars represent the standard deviation of four biological replicates, and the point represents the mean. (D) The *gal4Δ* deletion mutants showed a low ECAR phenotype compared to the parent (WT) control. (E) The *gal4Δ* deletion mutants displayed higher oxygen consumption rates (OCR) at 90 min post-inoculation compared to the parent (WT) control. The center lines of boxplots represent the median, the bounds of the boxes represent the interquartile range, the whiskers represent the spread of the data, and each dot represents one of three biological replicates. The *P* values shown are the results of two-sided *t* tests compared to the wild-type control. Source data are available online for this figure.

decreases (Conant and Wolfe, 2007). These results provide mechanistic insights into how the transcriptional activation of glycolysis in *Saturnispora* evolved and suggest that the modeling predicting the importance of these three key glycolytic enzymes extends to other yeasts beyond *S. cerevisiae*. Yeasts are also known to show high interspecific variation across their promoters, thereby rewiring expression patterns that influence trait elaboration related

to broad biological processes, including metabolism (Siddiq and Wittkopp, 2022; Kuang et al, 2018) and mating (Tsong et al, 2006).

To confirm the direct influence of Gal4p on the rapid ECAR phenotype, we developed a transformation protocol for *Sat. dispora*. The deletion of *GAL4* decreases the glucose consumption rate and renders the mutants more respiratory than fermentative. This result confirms the importance of the Gal4p transcription

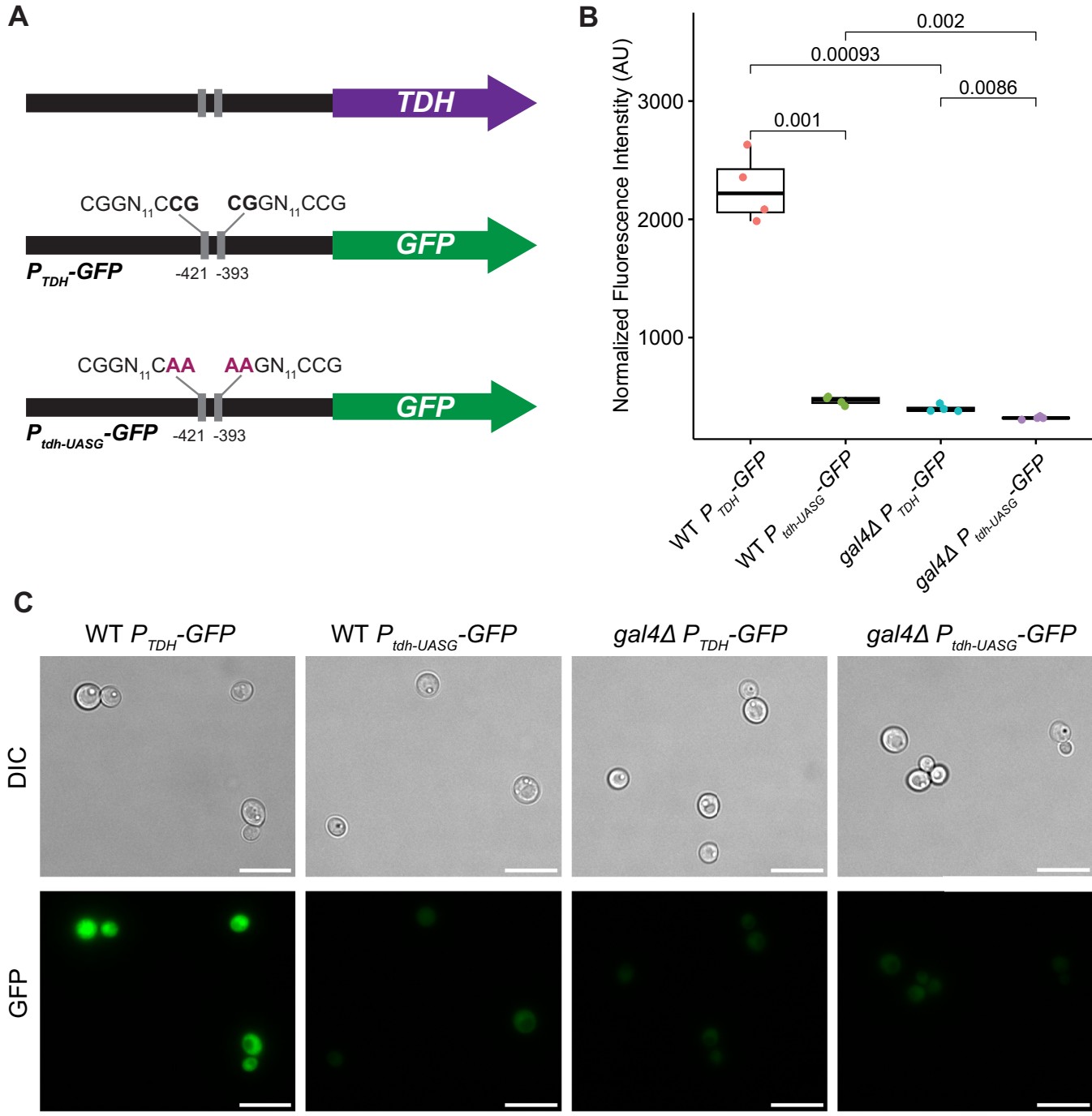

**Figure 5. Gal4p is required for the induction of the *TDH* promoter.**

(A) A schematic diagram showing the location of the predicted Gal4p-binding sites in the *TDH* promoter, the reporter construct, and the mutations made in the predicted Gal4p-binding sites in the reporter construct promoter. (B) The fluorescence intensity of the wild-type and *gal4Δ* deletion mutants containing the reporter with the wild-type *TDH* promoter ($P_{TDH}$-*GFP*) or the mutated *TDH* promoter ($P_{tdh-UASG}$-*GFP*). The fluorescence of each reporter-expressing strain was normalized to its respective control strain (i.e., wild-type and *gal4Δ* mutants) lacking any reporter. The center lines of boxplots represent the median, the bounds of the boxes represent the interquartile range, the whiskers represent the spread of the data, and each dot represents one of four biological replicates. The *P* values shown are the results of two-sided *t* tests comparing the strains as indicated. (C) Representative images of the GFP fluorescence from the strains carrying the reporter constructs. The scale bar represents 10 μm. Uncropped images are available as source data. Source data are available online for this figure.

factor in the rapid ECAR phenotype. To ensure that *GAL4* deletion had a specific impact on the cis-regulation of glycolytic genes, we developed a reporter system using the promoter of *TDH*, which was the most highly expressed gene in both *Sat. dispora* and *Sat. hagleri*. Furthermore, the enzyme encoded by *TDH* is modeled to be the rate-limiting step in glycolysis during the Crabtree/Warburg Effect in cancer cells (Shestov et al, 2014), which underscores its importance in aerobic fermentation. Through monitoring the expression of GFP as a reporter under the *TDH* promoter, we found that, if either the *GAL4* gene was missing or if the Gal4p-binding sites were mutated, the GFP expression was greatly reduced, showing that both the transcription factor Gal4p and its cis-regulatory binding sites are directly required for induction. Although Gal4p is required for the high glucose consumption rate of *Sat. dispora*, we can also infer that there are likely additional factors that contribute to the divergence in phenotypes of the rapid and low ECAR species. For example, the *gal4Δ* mutants still accumulated high levels of acetate and did not grow to the high biomass seen in the low ECAR species.

The yeast phenotype of overflow metabolism is one that, on the surface, seems wasteful as the organism produces fewer ATP molecules per glucose consumed. However, the rate of ATP production during fermentation is faster than the rate of ATP production during respiration, which potentially provides an additional evolutionary advantage (Liberti and Locasale, 2016). Furthermore, this strategy has been proposed to deplete a limited resource (glucose) and convert it into another product (ethanol), which would limit the growth of other microbes and could later be consumed. This "make-accumulate-consume" phenotype has already been observed in two distinct yeast lineages (Rozpedowska et al, 2011), and we now have evidence for a third independent lineage with a novel mechanism driven by increased transcriptional activation of glycolysis. There are also interesting parallels in the ecology of the rapid ECAR *Saturnispora* species and *S. cerevisiae*. *Sat. dispora* has primarily been isolated from tree exudates, and *Sat. hagleri* has been isolated from fruits and insects (Kurtzman et al, 2011; Harrison et al, 2024). Similar to *S. cerevisiae*, these yeasts are isolated from environments that have occasional periods of rich sugar availability, which may have prompted the evolution of rapid glycolysis (Liti, 2015; Jouhten et al, 2016). Although the rapid ECAR phenotype of *Sat. dispora* arose from a distinct cis-regulatory mechanism, many of the changes appear to lead to similar functional outcomes to those observed in *S. cerevisiae*. Even though the precise molecular mechanism of evolution differed between *Sat. dispora* and *S. cerevisiae*, the end result was still an increased abundance of lower glycolytic enzymes, glyceraldehyde-3-phosphate dehydrogenase, and hexokinase. Like the evolution of C4 photosynthesis in plants, the aerobic fermentation phenotype in yeasts represents another case of convergent evolution in metabolic efficiency.

In the case of C4 plants, the abundance, activity, or localization of the same key C4 enzymes has been optimized through convergent evolution, whereas we find that, in aerobically fermenting yeasts, the abundance of the same key glycolytic enzymes has been convergently increased through gene duplications or through the recruitment of cis-regulatory elements. This study provides new evidence supporting recent findings that trait evolution often employs the same genes across divergent lineages of yeasts (David et al, 2025; Aranguiz et al, 2025). While previous studies have focused on gene presence and family size, we were able to identify a mechanism of gene upregulation through the recruitment of cis-regulatory elements in the promoters of

congeneric species. These convergent adaptations suggest that there are likely strong constraints on how a yeast can evolve a rapid glycolytic rate. Indeed, high glycolytic rates and retained glycolytic duplicates may be a predictable occurrence, as suggested by multiple recently discovered novel yeast WGDs (David et al, 2026). Whether by gene duplication or cis-regulatory evolution, it seems that an increase in the abundance of the same set of glycolytic enzymes is required for yeasts to achieve high glycolytic rates. While organisms may deploy divergent molecular mechanisms due to chance and the historical contingencies of their lineages, these mounting examples suggest that evolutionary convergence operates within the constraints of the pathway being manipulated by natural selection.

# Methods

### Reagents and tools table

| Reagent/resource | Reference or source | Identifier or catalog number |
|---|---|---|
| **Experimental models** | | |
| Budding yeasts | Opulente et al, 2024 | Dataset EV1 |
| **Recombinant DNA** | | |
| pHLCH070 | JGI | This study (GenBank accession PX666697) |
| **Oligonucleotides and other sequence-based reagents** | | |
| Primer list | IDT | Appendix Table S1 |
| pRS426 | Christianson et al,1992 | |
| pUG6 | Guldener et al, 1996 | |
| pAG25 | Goldstein and McCusker, 1999 | |
| **Chemicals, enzymes, and other reagents** | | |
| Synthetic complete media | US Biologicals | Y2025 |
| Dropout mix | US Biologicals | D9515 |
| Ammonium sulfate | US Biologicals | 1450 |
| D-glucose | US Biologicals | G3050 |
| Phenol red | Sigma | P3532 |
| Acidic phenol | Sigma | 4682400 |
| RQ1 DNase | Promega | 6101 |
| NEBuilder® HiFi DNA Assembly | NEB | 2621 |
| *E. coli* 10 G Chemically Competent Cells | Lucigen | 601072 |
| Sorbitol | Sigma | 18765 |
| G418 | US Biologicals | 1000 |
| nourseothricin | WernerBioagents | 5.005 |
| Phusion polymerase | NEB | 530 |
| **Software** | | |
| iTOL v7 | Letunic and Bork, 2024 | |
| R v 4.2.2 | | |
| ggpubr | Kassambara, 2023 | |

| Reagent/resource | Reference or source | Identifier or catalog number |
|---|---|---|
| caper v1.0.3 | Orme, 2013 | |
| corrplot v0.95 | Wei and Simko, 2024 | |
| ggplot2 v 3.5.2 | Wickham, 2016 | |
| Trimmomatic v0.30 | Bolger et al, 2014 | |
| FastQC | Andrews 2010 | |
| HiSat v2.1.0 | Kim et al, 2019 | |
| HTSeq-count v2.0.3 | Putri et al, 2022 | |
| OrthoFinder | Emms and Kelly, 2019 | |
| Custom script | https://github.com/Linda-Horianopoulos/Aerobic_fermentation_scripts | |
| EdgeR | Robinson et al, 2009 | |
| Gffread v 0.12.7 | Pertea and Pertea, 2020 | |
| Yeastract | Teixeira et al, 2023 | |
| MAFFT v 7.222 | Katoh and Standley, 2013 | |
| Trimal v 1.4.1 | Capella-Gutierrez et al, 2009 | |
| IQTree v1.6.8 | Nguyen et al, 2015 | |
| **Other** | | |
| BMG Labtech Fluostar | BMG | |
| RNeasy kit | Qiagen | |
| NEBNext Poly(A) mRNA Magnetic Isolation Module | NEB | |
| NEBNext Ultra II Directional RNA Library Kit | NEB | |
| eporator | Eppendorf | |
| Resipher | Lucid Scientific | |
| EVOS FL microscope | ThermoScientific | |

## Strain and culture conditions

All strains screened were previously described (Shen et al, 2018; Opulente et al, 2024) and are listed in Dataset EV1. Yeasts were routinely maintained on yeast extract peptone dextrose (YPD) agar at room temperature. Fermentation and extracellular acidification assays were conducted using synthetic complete (SCD) media containing yeast nitrogen base, 5 g/L ammonium sulfate, complete dropout mix, and 20 g/L D-glucose (US Biologicals). All growth conditions were at room temperature (RT, 22 °C) unless otherwise specified.

## Extracellular acidification rate assay

All yeasts were grown for 24 h in 2 ml YPD, sub-cultured 1:4 in fresh SCD pH 6.8, and grown for an additional 6 h. These actively growing cells were harvested by centrifugation, washed twice in sterile water to remove excess media, and normalized by optical density. Cells were inoculated into a 96-well plate at $OD_{600nm} = 0.15$ in SCD supplemented with 0.002% phenol red. For each

strain, 2% dextrose or a water control was added immediately before beginning the assay. The absorbance at 560 nm was measured every 3 min for 90 min, and the change in extracellular pH was calculated based on a standard curve (Fig. EV1). The dextrose-dependent acidification rate was determined as the difference between the acidification rate in the presence of 2% dextrose and the acidification rate in the water control. For each yeast species, three biological replicates were conducted on different days in random order. The average ECAR was plotted as a color strip outside the phylogeny using iTOL v7 (Letunic and Bork, 2024). Growth and metabolic parameters from several of these yeasts were retrieved from a previous study (Hagman et al, 2013) and compared to measured ECAR values. Specifically, Pearson's correlation tests were used to look for the association between ECAR and these parameters and visualized using $R$ v 4.2.2 and the ggpubr v0.6.0 package (Kassambara, 2023).

## Comparative genomics

The orthogroups corresponding to genes encoding glycolytic enzymes were retrieved from the published OrthoFinder analysis (Opulente et al, 2024). A correlation test between the number of orthologs and the dextrose-dependent acidification of each species was conducted for each gene, as well as the total sum of glycolytic genes. This analysis was also repeated for the orders Saccharomycetales and Pichiales. To account for phylogeny, we used phylogenetic generalized least squares (PGLS) using $R$ v4.2.2 and the package caper v1.0.3 (Orme, 2013). Data were visualized using ggpubr v0.6.0 (Kassambara, 2023) and corrplot v0.95 (Wei and Simko, 2024).

## Growth and metabolite profiling

Yeast strains were grown overnight in YPD in biological quadruplicate. Overnight cultures were collected, washed twice in sterile water, and inoculated in 50 ml fresh SCD pH 6.8 in 250-ml baffled flasks at a density of $OD_{600nm} = 0.05$. The flasks were incubated at RT while shaking at 225 rpm. At each time point, $OD_{600nm}$ was measured, and the supernatant was analyzed for glucose, ethanol, and acetate concentrations using high-performance liquid chromatography with a refractive index detector (HPLC-RID), as previously described (Schwalbach et al, 2012). The mean and standard deviation across replicates were calculated in $R$ v4.2.2, and plots were made using the package ggplot2 v3.5.2 (Wickham, 2016).

## RNA extraction and sequencing

RNA was collected from log-phase cells using acidic phenol as previously described for yeast (Collart and Oliviero, 1993). Briefly, cells were collected and mixed with 0.1 volumes of 5% acid phenol in 95% ethanol, harvested by centrifugation, and flash frozen. Cell pellets were lysed with glass beads, and RNA was extracted with acidic phenol twice, followed by a final chloroform extraction. The RNA was treated with RQ1 DNase (Promega) and subsequently cleaned up using an RNeasy column (Qiagen). Since the rapid and low ECAR yeasts grew at different rates, to obtain mid-log phase for all strains, the rapid ECAR yeasts were collected at 15 h post-inoculation, and the low ECAR yeasts were collected at 18 h post-inoculation.

We enriched for mRNA using the NEBNext Poly(A) mRNA Magnetic Isolation Module (NEB), and libraries were prepared using the NEBNext Ultra II Directional RNA Library Kit for Illumina (NEB) following the manufacturer's instructions. Paired-end libraries were sequenced on a NovaSeq X Plus (Illumina) with $2 \times 150$ bp reads. Reads were trimmed using Trimmomatic v0.30 (Bolger et al, 2014) to remove adapters and subsequently checked in FastQC (Andrews, 2010). Reads were mapped to the respective genomes of each species using HiSat v2.1.0 (Kim et al, 2019). HTSeq-count v2.0.3 (Putri et al, 2022) was used to generate the counts of annotated genes using the non-unique fraction setting to avoid overcounting or discarding highly similar genes. To compare the transcriptome between species, we used OrthoFinder (Emms and Kelly, 2019) to identify groups of orthologous genes within the *Saturnispora* species. Subsequently, the sum of counts and gene lengths for each orthogroup was calculated using a custom Python script (https://github.com/Linda-Horianopoulos/Aerobic_fermentation_scripts). The counts and total gene lengths were then used to determine the TPM for each orthogroup in each species using EdgeR (Robinson et al, 2009). The orthogroups encoding glycolytic and TCA cycle enzymes were retrieved, and plots were made to visualize the expression in each species in *R* v4.2.2 using ggplot2 v3.5.2 (Wickham, 2016).

## Identification of putative transcription factor binding sites

Putative promoter sequences for each of the glycolytic genes were retrieved from the annotation files using gffread v0.12.7 (Pertea and Pertea, 2020). Yeastract (Teixeira et al, 2023) was used to identify all conserved *S. cerevisiae* transcription factor binding sites in the 1000 bp upstream of the coding sequence of each glycolytic gene from *Sat. dispora, Sat. hagleri, Sat. mendoncae, Sat. silvae, Martiniozyma abietophilia, P. kudriavzevii,* and *K. phaffii.* The number of predicted Gal4p-binding sites (exact consensus matches to CGGN$_{11}$CCG (Johnston, 1987)) in the promoter of each gene for all Pichiales species was determined by collecting all promoter sequences and using Yeastract to identify predicted Gal4p-binding sites. In the case that more than one ortholog was present, the ortholog with the highest number of binding sites in the promoter was visualized using iTOL v7 (Letunic and Bork, 2024).

## Phylogenetic analysis of *GAL4* orthologs

The genes in the same orthogroup as the canonical *S. cerevisiae* GAL4 (YPL248C) were retrieved for all species included in our ECAR screen. The translated amino acid sequences were retrieved, aligned using MAFFT v7.222 (Katoh and Standley, 2013), and the alignment was trimmed using trimal v1.4.1, applying the gappyout option (Capella-Gutiérrez et al, 2009). The phylogeny was inferred using IQtree v1.6.8 with automatic substitution model selection and 1000 bootstraps (Nguyen et al, 2015). The phylogeny was visualized in iTOL v7 (Letunic and Bork, 2024) and manually pruned to remove an outgroup clade on a disproportionately long internal branch, which encodes divergent C6 zinc transcription factors.

## Gene deletion

The coding sequence of *GAL4* in *Sat. dispora* was replaced with a *kanMX* marker using an electroporation protocol adapted from *K.*

*phaffii* (De Schutter and Callewaert, 2012). Briefly, ~1-kb regions upstream and downstream of the *GAL4* coding sequence were amplified from genomic DNA using primers with homology to the *kanMX* marker and the pRS426 backbone (Appendix Table S1). The *kanMX* marker was amplified from pUG6 (Güldener et al, 1996) and the pRS426 backbone (Christianson et al, 1992) was also amplified using primers specified in Appendix Table S1. These fragments were assembled using HiFi assembly (NEB) and transformed into *E. cloni* (Lucigen). The full construct was subsequently amplified from the plasmid, and 1 µg was transformed into *Sat. dispora*. Electroporation was performed in 80 µL of ice-cold 1 M sorbitol using 1.5 kV. After electroporation, cells were recovered in 1 mL of YPD + 1 M sorbitol and plated on YPD plates supplemented with 1 M sorbitol and 300 µg/mL G418. Transformants were screened for stable resistance against G418, and correct replacement of the coding sequence was confirmed with colony PCR. To allow the *kanMX* marker to be used for additional modifications, the *kanMX* cassette was replaced with a *natMX* cassette, which was amplified from pAG25 (Goldstein and McCusker, 1999), using the homology of the promoter and terminator. For selection of colonies integrating the *natMX* marker, colonies were plated on YPD plates supplemented with 1 M sorbitol and 25 µg/ml nourseothricin. To ensure that the phenotype was not due to any spurious background mutations, two independent colonies were used in all experiments. The strains constructed for this study are listed in Appendix Table S2.

## Oxygen consumption measurements

The Resipher system (Lucid Scientific) was used to monitor the oxygen levels and calculate the oxygen consumption rates (OCR). Three biological replicates of overnight cultures were washed twice in sterile water and normalized to OD600 = 0.1 in 95 µl in a 96-well plate in SC medium lacking a carbon source. The plate was centrifuged at $300 \times g$ for 2 min to pellet the cells to the bottom of the plate, and immediately before beginning reading, 5 µl of 40% dextrose was added for a final concentration of 2% dextrose. The plate was incubated in the dark at room temperature while the oxygen concentration was measured. The oxygen consumption rates were retrieved at the 90-min timepoint for consistency with the ECAR assays. The data was visualized in *R* using ggpubr v0.6.0 (Kassambara, 2023), and the significances of differences between strains were determined using two-sided *t* tests.

## Evaluating the cis-regulatory elements in the promoter of *TDH*

An insertion construct was designed in a genomic safe haven to allow heterologous proteins to be expressed from a region of the genome with no risk of gene disruption. A plasmid containing ~1 kb regions to direct homologous recombination to the genomic safe haven site flanking the sequence for GFP driven by the *TDH* promoter from *Sat. dispora* was synthesized by the Joint Genome Institute (JGI). Since this promoter had two predicted Gal4p-binding sites, we mutated them both using site-directed mutagenesis as previously described (Zheng et al, 2004) with the modification that Phusion polymerase was used to amplify the construct. Binding site mutants were confirmed by whole plasmid sequencing (Plasmidsaurus). Both versions of the reporter

construct were transformed into the *Sat. dispora* wild-type strain and *gal4Δ::natMX* deletion mutant using electroporation. Transformants were screened using colony PCR. Four biological replicates of each strain, including controls lacking any reporter, were grown overnight in YPD, washed twice in sterile water, and inoculated in fresh SC + D pH 6.8 at $OD_{600nm} = 0.1$. After growing for 40 h, cells were normalized to $OD_{600nm} = 1$ and the fluorescence intensity was measured using a BMG Labtech FLUOstar plate reader. The background from control strains without a reporter was subtracted, and the fluorescence intensity was plotted in *R* using the package ggpubr v0.6.0 (Kassambara, 2023). The statistical significance was determined using two-sided *t* tests.

Images of cells expressing GFP in the WT and *gal4Δ* background with the intact and mutated promoter grown under the same conditions (SC + D pH 6.8 for 40 h) were captured with an EVOS FL microscope with a ×100 oil immersion plan apochromat objective lens. Prior to imaging, samples were randomly coded to blind the researcher and prevent any bias in imaging. Images were uniformly processed and cropped in FIJI (Schindelin et al, 2012). Unedited images are available as source data.

## Data availability

The RNA-Sequencing data have been deposited to NCBI's GEO with the accession number GSE307312. All other data are provided as datasets or source data. Custom code is available on Github (https://github.com/Linda-Horianopoulos/Aerobic_fermentation_scripts) or as a release on Zenodo (https://doi.org/10.5281/zenodo.17185390). Upon request to the corresponding authors, all engineered strains are available under the Uniform Biological Material Transfer Agreement or another mutually agreeable material transfer agreement.

The source data of this paper are collected in the following database record: biostudies:S-SCDT-10_1038-S44318-026-00778-0.

## Peer review information

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

## Acknowledgements

The authors thank members of the Hittinger Lab and Y1000+ Project (http://y1000plus.org) team members for helpful discussions and the University of Wisconsin Biotechnology Center DNA Sequencing Facility (Research Resource Identifier, RRID:SCR_017759) for providing sequencing core services. This project was supported by the National Science Foundation under Grants No. DEB-2110403 (to CTH); DEB-2110404 (to AR); in part by the Great Lakes Bioenergy Research Center, U.S. Department of Energy, Office of Science, Biological and Environmental Research Program under Award Number DE-SC0018409 (of which CTH is a co-investigator); and the National Institute of Food and Agriculture, United States Department of Agriculture, Hatch project 7005101 (to CTH). Research in AR's lab is also supported by the National Institutes of Health/National Institute of Allergy and Infectious Diseases (R01 AI153356). The work (proposal: https://doi.org/10.46936/10.25585/60008615) conducted by the U.S. Department of Energy Joint Genome Institute (https://ror.org/04xm1d337), a DOE Office of Science User Facility, is supported by the Office of Science of the U.S. Department of Energy, operated under Contract No. DE-AC02-05CH11231. We specifically thank Ian Blaby for helping design and Robert Evans for synthesizing the safe haven insertion plasmid as part of Yasuo Yoshikuni's team. LCH was supported by a Natural Sciences and Engineering Research Council of Canada (NSERC) postdoctoral fellowship.

## Author contributions

**Linda C Horianopoulos**: Conceptualization; Formal analysis; Funding acquisition; Investigation; Visualization; Methodology; Writing—original draft. **Antonis Rokas**: Funding acquisition; Writing—review and editing. **Chris Todd Hittinger**: Conceptualization; Supervision; Funding acquisition; Writing—original draft; Project administration; Writing—review and editing.

Source data underlying figure panels in this paper may have individual authorship assigned. Where available, figure panel/source data authorship is listed in the following database record: biostudies:S-SCDT-10_1038-S44318-026-00778-0.

## Disclosure and competing interests statement

AR is a scientific consultant of LifeMine Therapeutics, Inc. The remaining authors declare no competing interests.

# Expanded View Figures

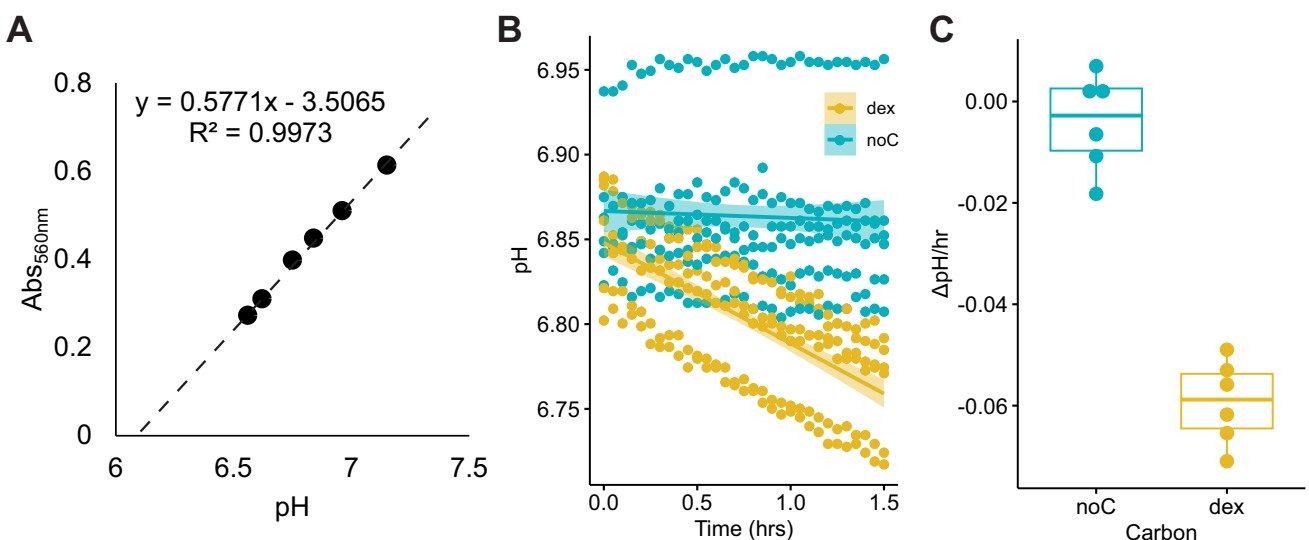

**Figure EV1. Optimization of the extracellular acidification rate (ECAR) assay.**

(**A**) A standard curve of the absorbance at 560 nm generated from media between pH 6.5 and 7.5. (**B**) Optimization assay using *S. cerevisiae* showing that the control with no carbon source (noC) did not acidify the media, whereas the cells with 2% glucose added (dex) consistently acidified the media across six biological replicates. (**C**) The slopes calculated from the six biological replicates showing the dextrose-dependent pH change. The center lines of boxplots represent the median, the bounds of the boxes represent the interquartile range, the whiskers represent the spread of the data, and each dot represents one of six biological replicates.

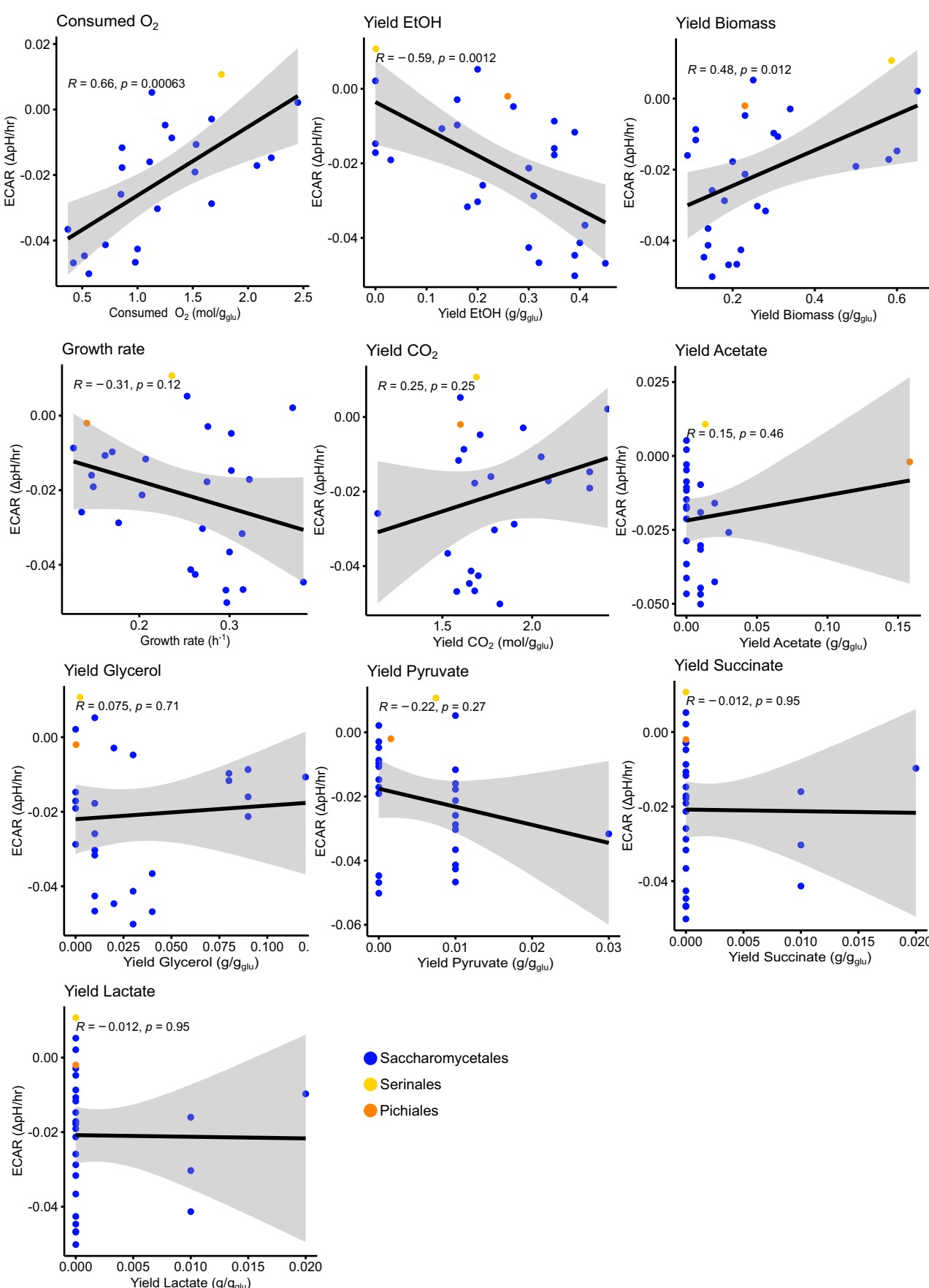

◀ **Figure EV2. Comparison of ECAR and parameters associated with the Crabtree/Warburg effect.**

The measured ECAR values were correlated with the parameters associated with ECAR collected by (Hagman et al, 2013). The R and *P* values are based on Pearson's correlations. Each dot represents a single species and is colored according to the taxonomic order of that species.

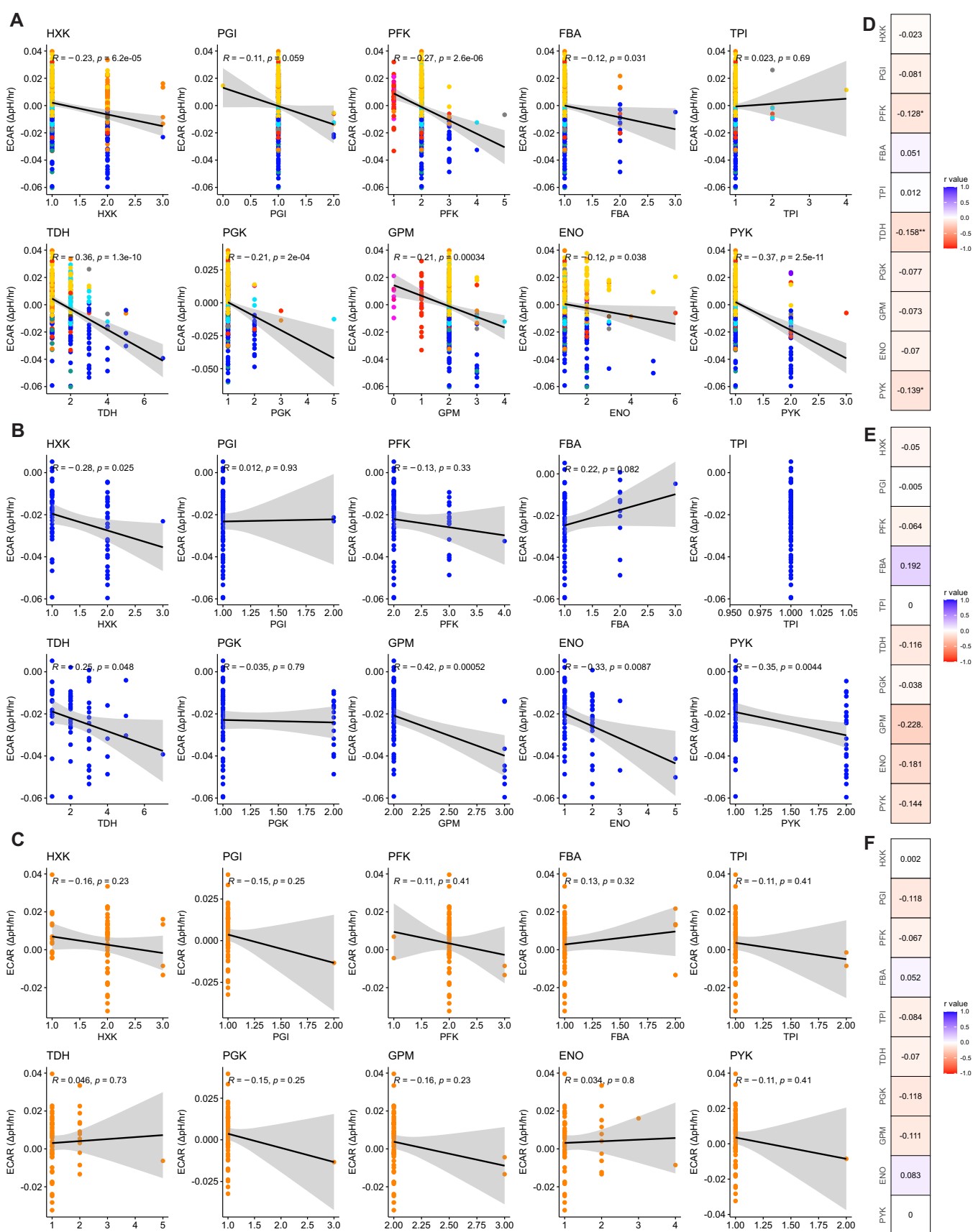

◀ **Figure EV3. The gene family size of glycolytic genes correlates with ECAR across all yeasts.**

(**A–C**) The correlation between ECAR and the number of homologs of each glycolytic gene showing the Pearson's correlation and $P$ value for (**A**) all phenotyped yeasts, (**B**) the yeasts in the order Saccharomycetales, and (**C**) the yeasts in the order Pichiales. (**D–F**) The phylogenetically corrected correlations between ECAR and the number of homologs of each glycolytic gene for (**D**) all phenotyped yeasts, (**E**) the yeasts in the order Saccharomycetales, and (**F**) the yeasts in the order Pichiales. The $R$ value from the phylogenetic generalized least squares is in the box and the $P$ value is indicated as *$P < 0.05$ and **$P < 0.01$. Data points are color-coded according to taxonomic order as in Fig. 1.

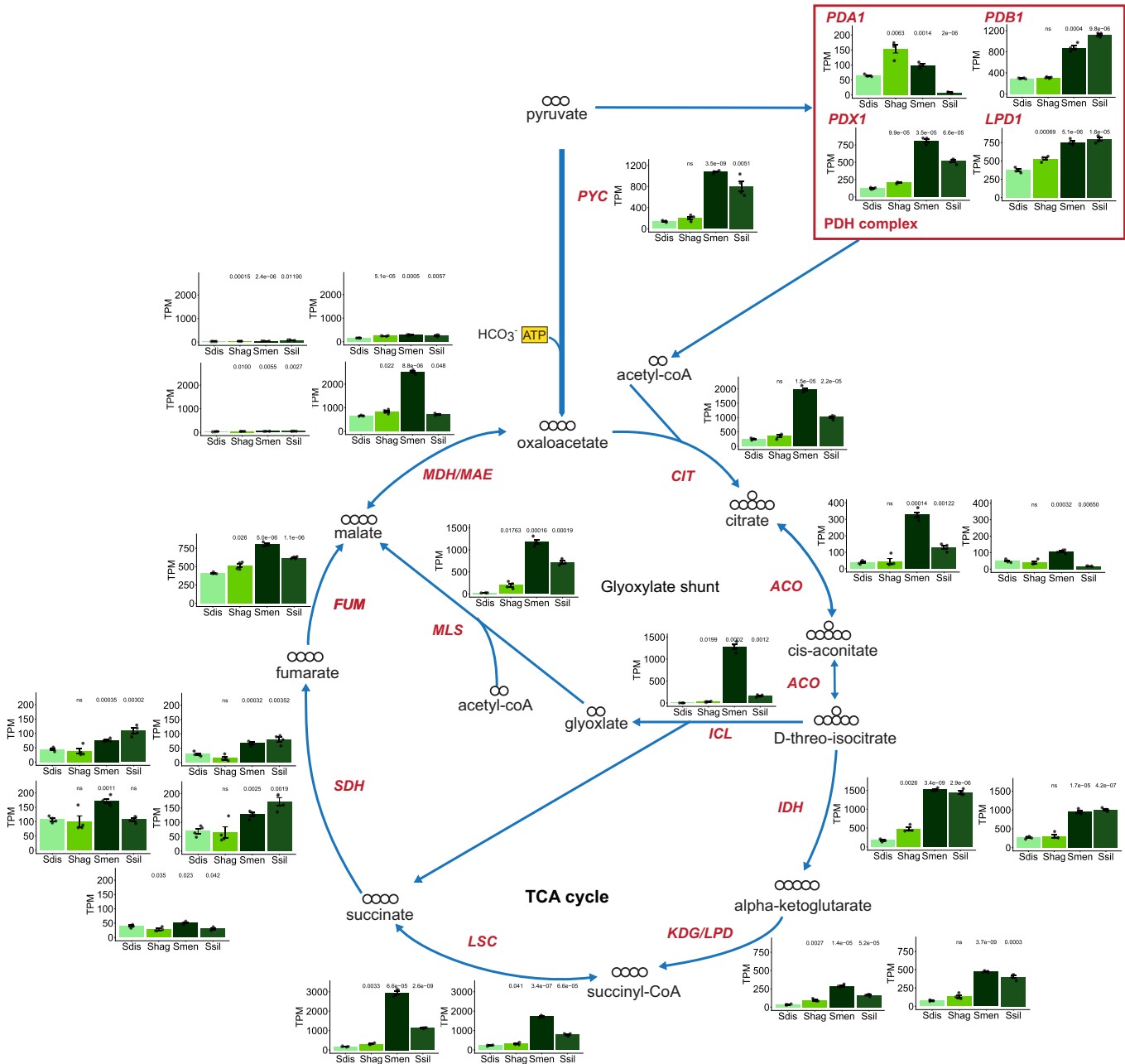

**Figure EV4. Divergent expression patterns of TCA cycle genes between rapid ECAR and low ECAR *Saturnispora* species.**

The gene expression for orthologous genes in the TCA cycle and glyoxylate shunt across two rapid ECAR (*Sat. dispora* Sdis, *Sat. hagleri* Shag) and two low ECAR (*Sat. mendoncae* Smen, *Sat. silvae* Ssil) are shown as transcripts per million (TPM). The bar height represents the average of four biological replicates: each replicate is shown as a black dot, and the error bars represent the standard error of the mean. The significance is relative to the TPM in *Sat. dispora* (ns not significant). In the schematic of glycolysis, carbon backbones are white.

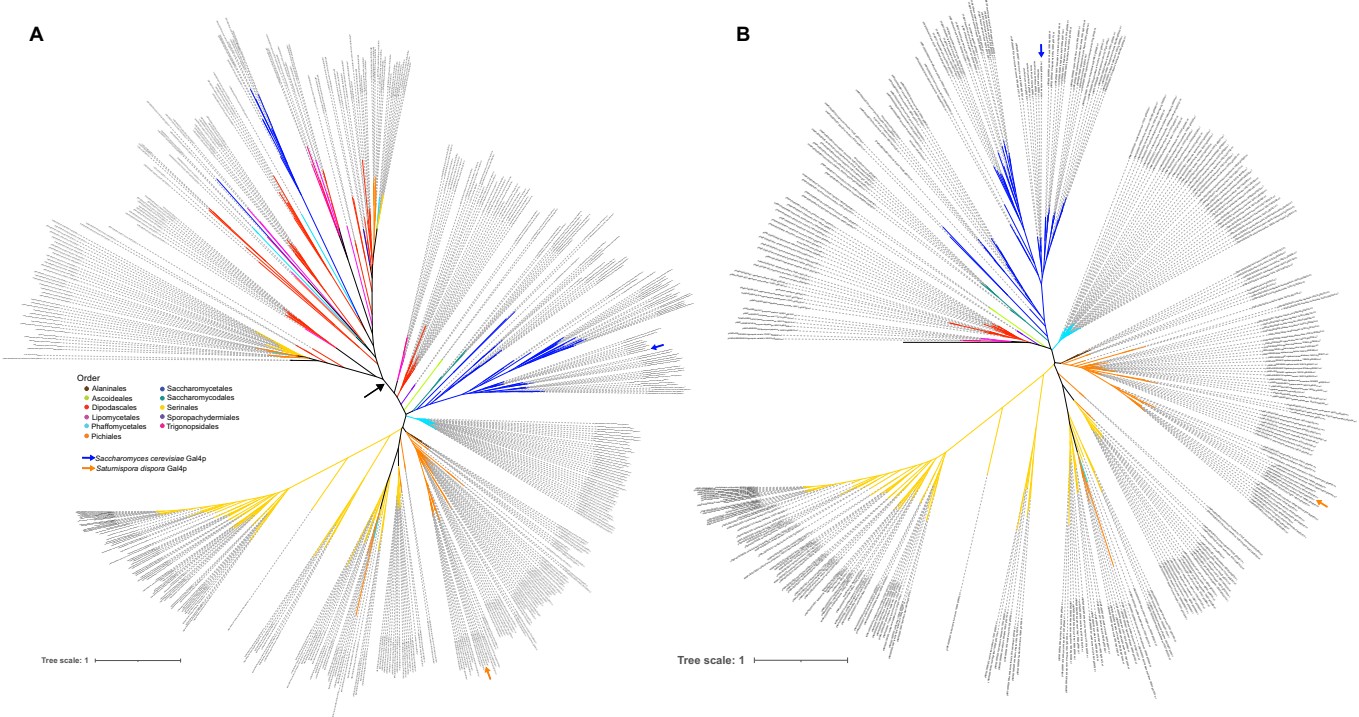

**Figure EV5.   Unrooted phylogeny of the Gal4p orthologs in the budding yeasts used in this study.**

(**A**) A phylogeny including all translated amino acid sequences of genes in the orthogroup containing canonical Gal4p. A black arrow indicates the location of a branch, which was pruned in (**B**) due to the length of the branch and the topological incongruence with the species phylogeny in this part of the tree, suggesting that these are likely other related C6 zinc transcription factors. The Newick files for these trees are available as source data. The location of the Gal4p orthologs from *S. cerevisiae* and *Sat. dispora* are highlighted on both trees, and the branches are colored according to the taxonomic order. Source data are available online for this figure.

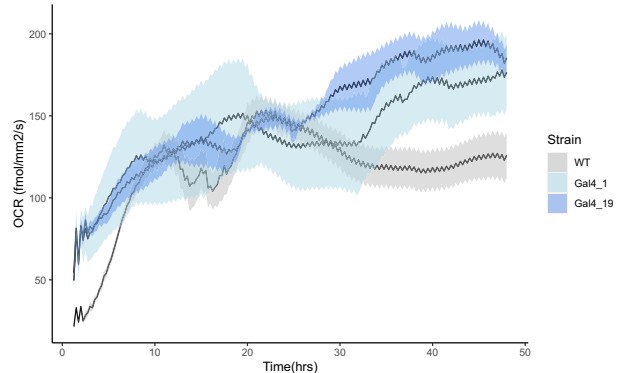

**Figure EV6.  The respiration profile of *Sat. dispora* and *gal4∆* deletion mutants over 48 h of growth.**

For each strain, the black line represents the average oxygen consumption rate (OCR), and the shaded area represents the standard error of the mean. Note that both mutants (Gal4_1, Gal4_19) have considerably higher OCR at several phases of the experiment, including the beginning and end.

