## [Peer Review File · The EMBO Journal]

Convergent evolution of aerobic fermentation through divergent mechanisms acting on key shared glycolytic genes

Linda Horianopoulos, Antonis Rokas, and Chris Hittinger

Corresponding authors: Chris Hittinger (chittinger@wisc.edu) , Linda Horianopoulos (lhoriano@uoguelph.ca)

Review Timeline:

Submission Date:	30th Sep 25
Editorial Decision:	28th Oct 25
Revision Received:	16th Dec 25
Editorial Decision:	27th Feb 26
Revision Received:	9th Mar 26
Accepted:	24th Mar 26

Editor: Yehu Moran

Transaction Report:

Dear Dr. Hittinger,

Thank you for submitting your manuscript for consideration by the EMBO Journal. It has now been seen by two referees whose comments are shown below.

Given the referees' positive recommendations, I would like to invite you to submit a revised version of the manuscript, addressing the comments of both reviewers. I should add that it is EMBO Journal policy to allow only a single round of major revision, and acceptance of your manuscript will therefore depend on the completeness of your responses in this revised version.

Thank you for the opportunity to consider your work for publication. I look forward to your revision.

Yours sincerely,

Yehu Moran
Academic Editor
The EMBO Journal

- a point-by-point response to the referees' comments, with a detailed description of the changes made (as a word file).
- a word file of the manuscript text.

- individual production quality figure files (one file per figure)

- a complete author checklist, which you can download from our author guidelines

(<https://www.embopress.org/page/journal/14602075/authorguide>).

- Expanded View files (replacing Supplementary Information)

- a Reagents and Tools Table as part of the Methods section, which can be downloaded from our author guidelines

(<https://www.embopress.org/page/journal/14602075/authorguide#structuredmethods>)

We realize that it is difficult to revise to a specific deadline. In the interest of protecting the conceptual advance provided by the work, we recommend a revision within 3 months (26th Jan 2026). Please discuss the revision progress ahead of this time with the editor if you require more time to complete the revisions.

Referee #1:

The study by Horianopoulos et al is an excellent piece of work that used an extracellular acidification assay to identify clades within the yeast genus *Saturnispora* that were distinguished by the rate of assimilation and metabolism of glucose. This was convincingly demonstrated to be due to increased expression of several genes of the glycolytic pathway. Interestingly, these are the same genes associated with the high glycolytic rate in the well-studied yeast *Saccharomyces cerevisiae*, though in that case the mechanism is related to increased copy number of the genes. In this study, detailed molecular work shows that the transcription factor Gal4 is responsible for the increased expression and high glycolytic flux. This work is of broad significance as it demonstrates independent (or convergent) evolution of mechanisms for rapid assimilation of glucose in diverse lineages. The authors suggest that similar constraints to those that applied for C4 plants may apply here and thus nature found a similar yet different way to achieve the same outcome.

The increased glycolytic flux is then linked to ethanol production under aerobic conditions and this is also a very significant topic. There are some points that need clarification or addressing to fully substantiate these conclusions. The main issue is that the rapid glycolytic flux is taken to be (almost) synonymous with aerobic respiration of sugars (i.e. the yeast Crabtree effect). It is generally accepted (and mentioned in the introduction) that the yeast Crabtree effect is mainly due to overflow metabolism when pyruvate cannot be sufficiently channelled to the TCA cycle and is diverted to acetaldehyde and then ethanol (and some acetate). As well as the capacity of the enzymes of overflow metabolism to accept the flux, the capacity of those of the TCA cycle is also important. It is known that respiratory capacity in the strongly Crabtree positive species *S. cerevisiae* is diminished and this is a contributory factor to its phenotype. Given the interlinked processes responsible for aerobic respiration, the article perhaps does not sufficiently interrogate its own datasets to strengthen the fundamental hypothesis put forward: that increased Gal4-mediated expression of specific glycolytic genes is primarily responsible for aerobic fermentation in this *Saturnispora* clade. I list below some areas where I think that this could be done to strengthen the conclusions

1. Figure 1 presents a graphical view of phylogeny and ECAR, taken to be correlated with rapid metabolism of glucose. The figure clearly shows the data that is key to the story of the paper - the divergence within the *Saturnispora* clade. But as this is also correlated in the text with aerobic fermentation (and demonstrated to be so for the tested *Saturnispora* species, it would be useful to have these data available in a supplementary excel file. I was also surprised not to see *K. phaffii* in the figure. One of the reasons for raising these points is because it appears that there are also some other (individual) species (e.g. *Blastobotrys nivea* and *Hansensula vinea*) that have comparable low ECAR values.
2. The degree of correlation with the Crabtree effect should also be mentioned - these data are available for most if not all of the species in figure 1 so this would be straightforward. In fact, it is likely to be a weak correlation as already it is evident that some Crabtree positive species (*Brettanomyces*) do not exhibit rapid ECAR. But this merits discussion.
3. Data on the alcohol dehydrogenase genes is not presented. It would be useful to know whether these genes are duplicated and how expression looks in the transcriptome data. One would expect high expression in the species that display aerobic fermentation (rapid ECAR) but the link with Gal4 would also be interesting. In the *gal4* mutant, ethanol production diminished but acetate production increased - that suggests an effect at the acetaldehyde branch?
4. The contribution of the TCA cycle to the overall phenomenon is not sufficiently considered. A supplementary figure of expression of these genes in the same format as figure 3 would help this discussion. Lower expression of TCA cycle genes also contributes to overflow metabolism and the relative levels of expression in the two *Saturnispora* clades merits some further analysis.

Other more minor points that could be addressed

5. It would be useful to see if the presence and number of Gal4 binding sites in other *Saturnispora* and closely related species to support the Gal4 hypothesis
6. It is not clear whether the analysis of promoters identified other putative TF binding sites - would be useful
7. It would be useful to explain to the general reader in the introduction why extracellular acidification (the mechanism) can be correlated with glucose assimilation as many readers will not be intimately familiar with yeast metabolism

Referee #2:

Review of Horianopoulos, Rokas and Hittinger, "Convergent evolution of aerobic fermentation through divergent mechanisms acting on key shared glycolytic genes."

Overview:

The authors use a high-throughput screen to test a very large number of yeast species for metabolic markers of high glucose flux that might be associated with aerobic fermentation. They find the expected relatives of *S. cerevisiae* to show these pattern, as well as another, distantly, related clade of yeasts. They verify the fermentative grow in the second clade and look for genomic differences associated with it. They find that regulatory changes to the expression of the enzymes of in the second half of glycolysis seem to have driven the phenotype in these other yeasts, in contrast to the apparent dosage-based source of fermentative growth in bakers yeast.

Major comments:

This is an excellent manuscript detailing a very interesting finding that has been carefully explored. My suggestions are more concerned with presentation.

- I think many readers will not instinctively understand how the acidification assay works, and how it is connected to glucose usage. I think the ECARS assay discussion on page 4 could be expanded by just a couple of sentences to help here.
- Another aspect of the rapid glucose use phenotype that could be mentioned is that the temporal dynamics of fermentation and respiration are known to be very different: The per-second rate at which fermentation produces ATP is enough higher than respiration that it may cancel out the efficiency advantages of respiration in terms of ATP/glucose molecule (Shestov, et al. 2014) (Liberti and Locasale 2016). The Crabtree effect thus may outcompete respiratory cells both because they consume resources faster and because they produce ATP faster.
- I don't if there's a possibility of further exploring the hexose transporters in these species, but the models suggest that these are actually often rate-limiting in glycolysis flux.

Minor point:

- The name *K. phaffii* is used in the text, but it looks like *Tetrapisispora phaffii* is the name used in the tree (assuming it is the same species being referred to). My understanding is that the latter is now more accepted, but I think just a consistent usage is sufficient.
- It might be work citing the older paper by Tsong et al., to reinforce the fact that yeasts apparently rewire their regulatory networks fairly easily (Tsong, et al. 2006).

References:

Liberti MV, Locasale JW. 2016. The Warburg effect: how does it benefit cancer cells? Trends in Biochemical Sciences 41:211-218.

Shestov AA, Liu X, Ser Z, Cluntun AA, Hung YP, Huang L, Kim D, Le A, Yellen G, Albeck JG. 2014. Quantitative determinants of aerobic glycolysis identify flux through the enzyme GAPDH as a limiting step. Elife 3:e03342.

Tsong AE, Tuch BB, Li H, Johnson AD. 2006. Evolution of alternative transcriptional circuits with identical logic. Nature 443:415-420.

-

Reviewer comments and responses

Referee #1:

The study by Horianopoulos et al is an excellent piece of work that used an extracellular acidification assay to identify clades within the yeast genus *Saturnispora* that were distinguished by the rate of assimilation and metabolism of glucose. This was convincingly demonstrated to be due to increased expression of several genes of the glycolytic pathway. Interestingly, these are the same genes associated with the high glycolytic rate in the well-studied yeast *Saccharomyces cerevisiae*, though in that case the mechanism is related to increased copy number of the genes. In this study, detailed molecular work shows that the transcription factor Gal4 is responsible for the increased expression and high glycolytic flux. This work is of broad significance as it demonstrates independent (or convergent) evolution of mechanisms for rapid assimilation of glucose in diverse lineages. The authors suggest that similar constraints to those that applied for C4 plants may apply here and thus nature found a similar yet different way to achieve the same outcome.

The increased glycolytic flux is then linked to ethanol production under aerobic conditions and this is also a very significant topic. There are some points that need clarification or addressing to fully substantiate these conclusions. The main issue is that the rapid glycolytic flux is taken to be (almost) synonymous with aerobic respiration of sugars (i.e. the yeast Crabtree effect). It is generally accepted (and mentioned in the introduction) that the yeast Crabtree effect is mainly due to overflow metabolism when pyruvate cannot be sufficiently channelled to the TCA cycle and is diverted to acetaldehyde and then ethanol (and some acetate). As well as the capacity of the enzymes of overflow metabolism to accept the flux, the capacity of those of the TCA cycle is also important. It is known that respiratory capacity in the strongly Crabtree positive species *S. cerevisiae* is diminished and this is a contributory factor to its phenotype. Given the interlinked processes responsible for aerobic respiration, the article perhaps does not sufficiently interrogate its own datasets to strengthen the fundamental hypothesis put forward: that increased Gal4-mediated expression of specific glycolytic genes is primarily responsible for aerobic fermentation in this *Saturnispora* clade. I list below some areas where I think that this could be done to strengthen the conclusions

We thank the reviewer for their appreciation and supportive comments about manuscript.

1. Figure 1 presents a graphical view of phylogeny and ECAR, taken to be correlated with rapid metabolism of glucose. The figure clearly shows the data that is key to the story of the paper - the divergence within the *Saturnispora* clade. But as this is also correlated in the text with aerobic fermentation (and demonstrated to be so for the tested *Saturnispora* species, it would be useful to have these data available in a supplementary excel file. I was also surprised not to see *K. phaffii* in the figure. One of the reasons for raising these points is because it appears that there are also some other (individual) species (e.g. *Blastobotrys nivea* and *Hansensula vinea*) that have comparable low ECAR values.

We thank the reviewer for the suggestion and apologize if supplemental tables were not made available. All ECAR values are reported in Dataset EV1, which also includes the number of orthologs of each glycolytic gene so that these figures can be reproduced or reanalyzed with more specific questions in mind. We agree that there are some additional species that show

interesting and rapid ECAR values, including *Blastobotrys niveus* and *Hanseniaspora vinea*. We hope that, by making these data publicly available, others with specific interests in these species can interrogate these data further.

The species which were included in the screen in this study were based on the species with genomes available from Shen et al. 2018, and *K. phaffii* was not included in that dataset; so unfortunately, we did not include it in our experimental design. Future work can focus on more species, particularly now that there are many more Pichiales with genomes sequenced (Opulente et al. 2024). Based on their genetic and phenotypic similarities, we also strongly suspect that *K. phaffii* would show a very similar phenotype to *K. pastoris*, which was included in the present study.

2. The degree of correlation with the Crabtree effect should also be mentioned - these data are available for most if not all of the species in figure 1 so this would be straightforward. In fact, it is likely to be a weak correlation as already it is evident that some Crabtree positive species (Brettanomyces) do not exhibit rapid ECAR. But this merits discussion.

We thank the reviewer for this suggestion as the Crabtree effect is certainly relevant to aerobic fermentation. We compared our ECAR data to several parameters measured by Hagman et al. in their 2013 paper, which is, to our knowledge, the most comprehensive and systematic analysis of the Crabtree effect across different yeast species. We find that the ECAR results are significantly correlated with oxygen consumption, ethanol yield, and biomass yield across these species, which are primarily in the order Saccharomycetales. Overall, we do expect that the ECAR phenotype would largely correlate with the Crabtree effect, but both datasets are phylogenetically skewed towards the Saccharomycetales, which makes it difficult to disentangle what is due to shared ancestry. If the reviewer is aware of a Crabtree effect dataset more comprehensive than Hagman et al. 2013, we would be happy to analyze it as well.

We have added an expanded view figure (now Figure EV2) to show these data and the following to the text lines 145-151:

“We also compared our ECAR data with parameters associated with the Crabtree/Warburg Effect using a previously collected dataset primarily composed of Saccharomycetales yeasts (Hagman *et al*, 2013). We found significant positive correlations between ECAR and the oxygen consumed as well as biomass yield, and a negative correlation between ECAR and ethanol yield (Fig. EV2, Dataset EV2). Since rapid acidification has a negative ECAR value, the rapid ECAR phenotype is thus associated with low oxygen consumption, low biomass yield, and high ethanol yield.”

Figure EV2. Comparison of ECAR and parameters associated with the Crabtree/Warburg Effect. The measured ECAR values were correlated with the parameters associated with ECAR collected by (Hagman et al, 2013). The R and p values are based on Pearson's correlations. Each dot represents a single species and is coloured according to the taxonomic order of that species.

3. Data on the alcohol dehydrogenase genes is not presented. It would be useful to know whether these genes are duplicated and how expression looks in the transcriptome data. One would expect high expression in the species that display aerobic fermentation (rapid ECAR) but the link with Gal4 would also be interesting. In the gal4 mutant, ethanol production diminished but acetate production increased - that suggests an effect at the acetaldehyde branch?

We thank the reviewer for this suggestion. Initially, we had decided not to include *PDC* and *ADH* genes because we were concerned that there may be genes with considerable sequence similarity that have divergent functions (e.g. thiamine biosynthesis genes and non-specific alcohol dehydrogenases). For the revised manuscript, we retrieved the orthogroups corresponding to *ADH* genes in our species, and although there is high variability in the gene family size across all yeasts, there was no variation within the *Saturnispora*. We have included these data on *ADH*, *PDC*, and *HXT* copy number in Dataset EV1. For *ADH* expression levels, we found a strange expression pattern with similarly high levels of *ADH* expression in the rapid ECAR *Sat. dispersa* and the low ECAR *Sat. silvae*. However, since different orthologs of alcohol dehydrogenase can catalyze either direction of the reaction between ethanol and acetaldehyde, we cannot be sure that this corresponds to more activity in the forward direction towards ethanol.

The other reviewer made a similar suggestion to include additional genes, and overall, the new results were quite informative. Thus, we have included the *ADH*, *PDC*, and *HXT* genes in our revised analyses regarding gene expression (Fig. 3) and promoter content (Fig. 4).

Figure 3. Divergent expression patterns of glycolytic genes between rapid ECAR and low ECAR *Saturnispora* species. The gene expression for orthologous glycolytic genes across two rapid ECAR (*Sat. dispora* = Sdis, *Sat. hagleri* = Shag) and two low ECAR (*Sat. mendoncae* = Smen, *Sat. silvae* = Ssil) are shown as transcripts per million (TPM). The bar height represents the average of four biological replicates, each replicate is shown as a black dot, and the error bars represent the standard error of the mean. The significance is relative to the TPM in *Sat. dispora* (ns = not significant, * $p < 0.05$, ** $p < 0.01$, *** $p < 0.005$, **** $p < 0.001$). In the schematic of glycolysis, carbon backbones are white, and phosphates are yellow. Boxes around gene names are orthologous to genes that have retained duplicates in *S. cerevisiae* as in Fig. 1C; solid boxes indicate duplicates retained in all post-WGD yeasts, whereas the dashed boxes indicate duplicates retained in some but not all post-WGD yeasts (Conant & Wolfe, 2007).

At the promoter level we see Gal4p-binding sites in promoters driving *ADH* expression in all *Saturnispora* species except for *Sat. mendoncae*. This observation is potentially consistent with its low ethanol production and low *ADH* expression. Since we do not know whether these orthologs are driving the forward or reverse reactions, it is hard to interpret these results too strongly, although it may show at least some degree of coordinated promoter rewiring across *Saturnispora* species.

We have updated Figure 4 to include the data on *ADH* and an expanded set of Pichiales species.

Figure 4. The Gal4p transcription factor contributes to the rapid ECAR metabolic phenotype in *Sat. dispersa*. (A) The number of predicted Gal4p-binding sites in the putative promoters 1kb upstream of each glycolytic gene in all tested Pichiales species. The genes marked with an asterisk have an additional binding site in the rapid ECAR species. The focal rapid and low ECAR species used in this study are highlighted with arrows on the phylogeny. The (B) growth and (C) extracellular metabolites for the *Sat. dispersa* wild-type (WT) control and two independent *gal4Δ* deletion mutants. Two high ECAR and two low ECAR species were measured during growth in a highly aerobic 250-ml baffled flask. The error bars represent the standard deviation of four biological replicates. (D) The *gal4Δ* deletion mutants showed a low ECAR phenotype compared to the parent (WT) control. (E) The *gal4Δ* deletion mutants displayed higher oxygen consumption rates (OCR) at 90 minutes post inoculation compared to the parent (WT) control. The centre lines of boxplots represent the median, the bounds of the boxes represent the interquartile range, the whiskers represent the spread of the data, and each dot represents one of three biological replicates. The p values shown are the results of two-sided t-tests compared to the wild-type control.

4. The contribution of the TCA cycle to the overall phenomenon is not sufficiently considered. A supplementary figure of expression of these genes in the same format as figure 3 would help this discussion. Lower expression of TCA cycle genes also contributes to overflow metabolism and the relative levels of expression in the two *Saturnispora* clades merits some further analysis.

We thank the reviewer for this helpful suggestion. We have generated an additional expanded view figure for the TCA cycle (Figure EV4). As the reviewer hypothesized, we see the reverse trend of what we saw in the glycolytic pathway: that is higher expression in the slow ECAR *Saturnispora* species.

We note that there are clear and significant differences in specific genes, including *PYC* and some genes encoding components of the pyruvate dehydrogenase complex, which may be a bottleneck to pyruvate entering the citric acid cycle.

We have incorporated these new results on lines 236-244:

“Accordingly, we also found that the low ECAR species had higher expression of genes encoding tricarboxylic acid (TCA) cycle enzymes. In particular, we noted that the expression of the gene encoding pyruvate carboxylase and multiple genes encoding components of the pyruvate dehydrogenase complex had significantly lower expression in the rapid ECAR species, which suggests that reduced respiratory capacity also contributes to overflow metabolism (Fig. EV4). The differences at the transcriptional level of both glycolysis and the TCA cycle support the conclusion that the metabolisms of these closely related yeasts are highly divergent and that low ECAR species favour aerobic respiration.”

Figure EV4. Divergent expression patterns of TCA cycle genes between rapid ECAR and low ECAR *Saturnispora* species. The gene expression for orthologous genes in the TCA cycle and glyoxylate shunt across two rapid ECAR (*Sat. disporsa* = Sdis, *Sat. hagleri* = Shag) and two low ECAR (*Sat. mendoncae* = Smen, *Sat. silvae* = Ssil) are shown as transcripts per million (TPM). The bar height represents the average of four biological replicates; each replicate is shown as a black dot, and the error bars represent the standard error of the mean. The significance is relative to the TPM in *Sat. disporsa* (ns = not significant, * $p < 0.05$, ** $p < 0.01$, *** $p < 0.005$, **** $p < 0.001$). In the schematic of glycolysis, carbon backbones are white.

Other more minor points that could be addressed

5. It would be useful to see if the presence and number of Gal4 binding sites in other *Saturnispora* and closely related species to support the Gal4 hypothesis

We thank the reviewer for this suggestion, which prompted us to interrogate the distribution further by looking for Gal4p-binding sites across all Pichiales species included in this study (Fig. 4 above). Although we were primarily interested in the difference between rapid and low ECAR *Saturnispora* species, this new analysis revealed that there are overall more Gal4p-binding sites in *Saturnispora* species compared to other species in the order, particularly in the promoters driving the expression of the genes encoding the key enzymes: *HXK*, *TDH*, and *ENO*. We have also made these data available in Dataset EV4.

6. It is not clear whether the analysis of promoters identified other putative TF binding sites - would be useful

Yes, the full list of predicted binding sites is provided in Dataset EV3 for the *Saturnispora* species. For the additional analysis of all Pichiales, we only identified Gal4p-binding sites which are provided in Dataset EV4.

7. It would be useful to explain to the general reader in the introduction why extracellular acidification (the mechanism) can be correlated with glucose assimilation as many readers will not be intimately familiar with yeast metabolism

Thank you for the suggestion. We have added a couple sentences to our introduction about the principle upon which the assay relies on lines 83-90:

“Acidification measurements have also been applied in *S. cerevisiae* to assess fermentative capacity in industrial settings (Sigler, 2013; Opekarova & Sigler, 1982; Yamashoji et al, 2020; Sigler et al, 1981). Acidification occurs in yeast during glucose assimilation as neutral glucose is taken up, phosphorylated, and converted into charged glycolytic intermediates, organic acids, and carbon dioxide. Collectively, these lower the pH of the media and the cytosol, but the cell must keep the cytosol buffered. Thus, the cell extrudes protons which rapidly lowers the extracellular pH upon glucose addition (Kotyk et al, 2003; Orij et al, 2011, 2009).”

Referee #2:

Review of Horianopoulos, Rokas and Hittinger, "Convergent evolution of aerobic fermentation through divergent mechanisms acting on key shared glycolytic genes."

Overview:

The authors use a high-throughput screen to test a very large number of yeast species for metabolic markers of high glucose flux that might be associated with aerobic fermentation. They find the expected relatives of *S. cerevisiae* to show these pattern, as well as another, distantly, related clade of yeasts. They verify the fermentative grow in the second clade and look for genomic differences associated with it. They find that regulatory changes to the expression of the enzymes of in the second half of glycolysis seem to have driven the phenotype in these other yeasts, in contrast to the apparent dosage-based source of fermentative growth in bakers

yeast.

Major comments:

This is an excellent manuscript detailing a very interesting finding that has been carefully explored. My suggestions are more concerned with presentation.

- I think many readers will not instinctively understand how the acidification assay works, and how it is connected to glucose usage. I think the ECARS assay discussion on page 4 could be expanded by just a couple of sentences to help here.

Thank you for this suggestion. The other reviewer also felt this would benefit from a better description. We have added a couple sentences to describe the general mechanism behind the ECAR assay to lines 83-90:

“Acidification measurements have also been applied in *S. cerevisiae* to assess fermentative capacity in industrial settings (Sigler, 2013; Opekarova & Sigler, 1982; Yamashoji et al, 2020; Sigler et al, 1981). Acidification occurs in yeast during glucose assimilation as neutral glucose is taken up, phosphorylated, and converted into charged glycolytic intermediates, organic acids, and carbon dioxide. Collectively, these lower the pH of the media and the cytosol, but the cell must keep the cytosol buffered. Thus, the cell extrudes protons which rapidly lowers the extracellular pH upon glucose addition (Kotyk et al, 2003; Orij et al, 2011, 2009)”

- Another aspect of the rapid glucose use phenotype that could be mentioned is that the temporal dynamics of fermentation and respiration are known to be very different: The per-second rate at which fermentation produces ATP is enough higher than respiration that it may cancel out the efficiency advantages of respiration in terms of ATP/glucose molecule (Shestov, et al. 2014) (Liberti and Locasale 2016). The Crabtree effect thus may outcompete respiratory cells both because they consume resources faster and because they produce ATP faster.

This is an excellent point, and we have adapted our explanation on the potential evolutionary advantages of this rapid glycolysis phenotype on lines 395-401:

“The yeast phenotype of overflow metabolism is one that, on the surface, seems wasteful as the organism produces fewer ATP molecules per glucose consumed. However, the rate of ATP production during fermentation is faster than the rate of ATP production during respiration which potentially provides an additional evolutionary advantage (Liberti & Locasale, 2016). Furthermore, this strategy has been proposed to deplete a limited resource (glucose) and convert it into another product (ethanol), which would limit the growth of other microbes and could later be consumed.”

The article by Shestov et al., was also very compelling in that it focuses on *GAPDH* (or as we call it in yeast, *TDH*) as the rate limiting step in aerobic glycolysis in cancer cells. Since we found a significant difference in *TDH* expression and an extra Gal4 binding site in its promoters in rapid ECAR yeasts, we cited this work in that section to show additional evidence of the importance of this step, lines 381-386:

“To ensure that *GAL4* deletion had a specific impact on the cis-regulation of glycolytic genes, we developed a reporter system using the promoter of *TDH*, which was the most highly expressed gene in both *Sat. dispersa* and *Sat. hagleri*. Furthermore, the enzyme encoded by *TDH* is

modelled to be the rate limiting step in glycolysis during the Crabtree/Warburg Effect in cancer cells (Shestov *et al*, 2014) which underscores its importance in aerobic fermentation.”

- I don't if there's a possibility of further exploring the hexose transporters in these species, but the models suggest that these are actually often rate-limiting in glycolysis flux.

Curiously, the hexose transporter gene family sizes are the opposite of what one might expect; specifically, the slow ECAR species have an additional transporter compared to the rapid ECAR species. This observation is the opposite of what is observed with the expansion of hexose transporters in the lineage leading to *S. cerevisiae*. These data have been included in Dataset EV1. However, the RNA-Seq data revealed that hexose transporters had higher expression in the rapid ECAR species, which is consistent with the reviewer's hypothesis. There are also conserved Gal4p-binding sites in the promoter of *HXT* genes in both rapid ECAR species. One of the slow ECAR species, *Sat. silvae*, also has a Gal4p-binding site in the promoter of one of its *HXT* genes; however, it is further upstream from the transcription start site (Dataset EV3).

We have added the hexose transporters to our transcriptome (Fig. 3) and Gal4p-binding site analyses (Fig 4).

We have also added the following statement to the Results section on lines 232-235: “The rapid ECAR species also had significantly higher expression of the genes encoding the hexose transporters (*HXT*) and the first step towards alcoholic fermentation, pyruvate decarboxylase (*PDC*) (Fig. 3).”

Minor point:

- The name *K. phaffii* is used in the text, but it looks like *Tetrapisispora phaffii* is the name used in the tree (assuming it is the same species being referred to). My understanding is that the latter is now more accepted, but I think just a consistent usage is sufficient.

We apologize for the lack of clarity; these are two different species. *K. phaffii* was not actually included in the screen since it was not in the Shen *et al*, 2018 dataset around which our experimental design was based. However, since it is one of the best studied Pichiales yeasts, we did want to include the relevant work on its metabolism in our Introduction and Discussion.

- It might be worth citing the older paper by Tsong *et al.*, to reinforce the fact that yeasts apparently rewire their regulatory networks fairly easily (Tsong, *et al.* 2006).

This addition is an excellent idea that has strengthened the framing of these results. We also believe that the regulatory rewiring model is further supported by our expanded systematic analysis of Gal4p-binding sites in the promoters of all Pichiales yeasts included in the revisions of this study (Fig. 4A). Through this analysis, we can see that gains and losses of these sites occur frequently during macroevolution.

We have added a sentence in the Discussion, lines: 374-377:

“Yeasts are also known to show high interspecific variation across their promoters thereby rewiring expression patterns that influence trait elaboration related to broad biological processes including metabolism (Siddiq & Wittkopp, 2022; Kuang *et al*, 2018) and mating (Tsong *et al*, 2006).”

References:

Liberti MV, Locasale JW. 2016. The Warburg effect: how does it benefit cancer cells? Trends in Biochemical Sciences 41:211-218.

Shestov AA, Liu X, Ser Z, Cluntun AA, Hung YP, Huang L, Kim D, Le A, Yellen G, Albeck JG. 2014. Quantitative determinants of aerobic glycolysis identify flux through the enzyme GAPDH as a limiting step. Elife 3:e03342.

Tsong AE, Tuch BB, Li H, Johnson AD. 2006. Evolution of alternative transcriptional circuits with identical logic. Nature 443:415-420.

-

Dear Dr. Hittinger,

Thank you for submitting your manuscript for consideration by the EMBO Journal. I would like to start by apologizing for the delay in getting back to you with a decision letter. This is mostly due to one of the original reviewers being very slow in their response. As you will see, one of the referee's still has some minor comments that require your attention. Moreover, our editorial assistance team flagged several issues that must be addressed before we can accept your manuscript for publication.

I would like to invite you to submit a revised version of the manuscript, addressing the comments.

When preparing your letter of response to the referee's comments, please bear in mind that this will form part of the Review Process File, and will therefore be available online to the community. For more details on our Transparent Editorial Process, please review our Editorial Policies page: <https://link.springer.com/partners/embo-press/editorial-policies>
Please note that the issues raised by the editorial assistance team must be corrected, but they do not need to be addressed in your response letter.

We generally allow three months as standard revision time. As a matter of policy, competing manuscripts published during this period will not negatively impact on our assessment of the conceptual advance presented by your study. Yet, in light of the minor nature of the comments I believe you would probably need a shorter time to complete your revision.

Thank you for the opportunity to consider your work for publication. I look forward to your revision.

Yours sincerely,

Yehu Moran
Academic Editor
The EMBO Journal

Read our guidance for manuscript revisions and related editorial policies: <https://link.springer.com/journal/44318/submission-guidelines#cms-Revised-submissions>

<https://media.springernature.com/original/springer-cms/rest/v1/content/27825798/data/v1>

- a point-by-point response to the referees' comments, with a detailed description of the changes made (as a word file).
- a word file of the manuscript text.
- individual production quality figure files (one file per figure)
- a complete author checklist
- Expanded View files (replacing Supplementary Information)
- a Reagents and Tools Table as part of the Methods section

Please remember: Digital image enhancement is acceptable practice, as long as it accurately represents the original data and conforms to community standards. If a figure has been subjected to significant electronic manipulation, this must be noted in the figure legend or in the 'Methods' section. The editors reserve the right to request original versions of figures and the original images that were used to assemble the figure.

We realize that it is difficult to revise to a specific deadline. In the interest of protecting the conceptual advance provided by the work, we recommend a revision within 3 months (28th May 2026). Please discuss the revision progress ahead of this time with the editor if you require more time to complete the revisions.

Comments by editorial assistance team

MANUSCRIPT FORMAT: Needs fixing. Figures/EV figures need to be removed from manuscript, only figure legends to be placed at the end of the manuscript text.

COI/DCIS: Disclosure and Competing Interests Statement title should be corrected.

AUTHORS: We detected name discrepancy: Linda Horianopoulos (system) vs Linda C. Horianopoulos (manuscript text); Chris Hittinger (system) vs Chris Todd Hittinger (manuscript). Please correct.

Author contributions/CRedit: should be removed from manuscript text and provided only in the system.

FUNDING INFO: "Please see Acknowledgments" sentence in the Comments box and one entry needs to be removed and all funders acknowledged in the manuscript need to be inserted as separate entries in the system.

Figure callouts in the text: missing for Figure 4B-D, Figure 5A-C; Reagents table has one wrong callout (missing an "S") - Appendix Table 1. Please correct.

DATASET EV LEGENDS: 4 datasets uploaded - Dataset EV1 and Dataset EV3 have wrong labels in their sheets (Table EV1 and Table EV3); the source file names for the datasets are not OK (Table EV# instead of Dataset EV#); dataset legends need to be removed from the manuscript file and inserted in each Excel file as a separate tab/sheet.

APPENDIX FILE WITH Table of contents: needs to be in PDF, needs a Table of Contents on the title page with each item listed with its page number

SYNOPSIS IMAGE: missing. Please provide.

SYNOPSIS TEXT: missing. Please provide.

SOURCE DATA (SD): SD files provided without the completed SD checklist so we can't check the SD; the SD need to be provided as folder per figure with clearly labeled panels

==>Update: SD checklist now provided and SD re-organized; there are also 2 folders for Figure 1 and 2 - we need these incorporated into the separate and existing folders of Figure 1 Source Data and Figure 2 Source Data; SD for 4C appears to be missing. Authors, please be in touch with the editorial office to fix this.

Figure Legends - Comments

- Please note that the exact p values are not provided in the legends of figures 3, EV4. This should be corrected.
- Please note that the box plots need to be defined in terms of minima, maxima, centre, bounds of box and whiskers, and percentile in the legends of figures 2A, EV1 C. Please correct.
- Please note that information related to n is missing in the legends of figures 2A. Please provide.
- Please note that the measure of center for the error bars needs to be defined in the legends of figures 2B, C; 4B, C". Please correct.

Comments by referees

Referee #1:

I commend the authors on a detailed response to the comments I made to the original manuscript. I carefully read the revised material, addressing the queries raised by both reviewers. I am fully satisfied with the responses, and with the changes that were made. This is an excellent piece of work that will be very well received by the research community.

I do have two minor suggestions to amend a couple of sentences that are changed/new in this version.

1. I don't think that the change to the final sentence of the abstract adds clarity. I think that the authors should revisit this again to make their point more coherently.

2. I was pleased to see the explanation of extracellular acidification (lines 89 - 93) but think that two things are conflated with the statement that "Collectively, these lower the pH of the media and the cytosol". I think that the main acidification is due to proton extrusion to control cytosolic pH, though the authors suggest that there is also a contribution also from secretion of other metabolites (CO₂, organic acids). If the authors want to say that there are two processes going on, they should make sure that they differentiate between them.

Referee #2:

The authors have addressed my concerns, and I have no further notes.

Reviewer comments and responses

Referee #1:

I commend the authors on a detailed response to the comments I made to the original manuscript. I carefully read the revised material, addressing the queries raised by both reviewers. I am fully satisfied with the responses, and with the changes that were made. This is an excellent piece of work that will be very well received by the research community.

We sincerely appreciate the supportive comments of the reviewer.

I do have two minor suggestions to amend a couple of sentences that are changed/new in this version.

1. I don't think that the change to the final sentence of the abstract adds clarity. I think that the authors should revisit this again to make their point more coherently.

We have adjusted this sentence and made it more specific to aerobic fermentation:

“These divergent genetic mechanisms affecting the same set of genes suggest that there are strong evolutionary constraints on how aerobic fermentation can arise.”

2. I was pleased to see the explanation of extracellular acidification (lines 89 - 93) but think that two things are conflated with the statement that "Collectively, these lower the pH of the media and the cytosol". I think that the main acidification is due to proton extrusion to control cytosolic pH, though the authors suggest that there is also a contribution also from secretion of other metabolites (CO₂, organic acids). If the authors want to say that there are two processes going on, they should make sure that they differentiate between them.

We agree that the major mechanism during these early timepoints is proton extrusion, but we wanted to acknowledge the potential contribution of the accumulation of organic acids in the media. It is a good idea to rephrase this statement for clarity:

“Acidification occurs in yeast during glucose assimilation as neutral glucose is taken up, phosphorylated, and converted into charged glycolytic intermediates, organic acids, and carbon dioxide. The intracellular accumulation of charged intermediates and organic acids lowers the pH of the cytosol, but the cell must keep the cytosol buffered to ensure proper enzymatic functions. Thus, the cell extrudes protons, which rapidly lowers the extracellular pH upon glucose addition (Kotyk et al, 2003; Orij et al, 2011, 2009). At later culture stages, the extracellular accumulation of organic acids can further lower the extracellular pH (Sigler & Hofer, 1991).”

Referee #2:

The authors have addressed my concerns, and I have no further notes.

We thank the reviewer for their support.

Dear Dr. Hittinger,

I am pleased to inform you that your manuscript has been accepted for publication in the EMBO Journal.

You may qualify for financial assistance for your publication charges - either via a Springer Nature fully open access agreement or an EMBO initiative. Check your eligibility: <https://link.springer.com/journal/44318/how-to-publish-with-us>

Yours sincerely,

Yehu Moran
Academic Editor
The EMBO Journal

Please note that it is The EMBO Journal policy for the transcript of the editorial process (containing referee reports and your response letters) to be published as an online supplement to each paper. If you should prefer removal of any referee-only figures included in the point-by-point response(s), e.g. because they may still be used for future publication or because they have been reproduced from published work by others, please do let us know immediately via response email.

More information is available here: <https://link.springer.com/partners/embo-press/editorial-policies#Peer%20review>